# Hierarchical Deconvolution for Incoherent Scatter Radar Data

Snizhana Ross[1], Arttu Arjas[2], Ilkka I. Virtanen[3], Mikko J. Sillanpää[4], Lassi Roininen[5], and Andreas Hauptmann[6]

[1,2,4,6]Research Unit of Mathematical Sciences, University of Oulu, FI-90014 Oulu, Finland
[3]Research Unit of Space Physics and Astronomy, University of Oulu, FI-90014 Oulu, Finland
[5]School of Engineering Science, Lappeenranta-Lahti University of Technology, FI-53851 Lappeenranta, Finland
[6]Department of Computer Science, University College London, London WC1E 6BT, UK

**Correspondence:** Snizhana Ross (Snizhana.Ross@oulu.fi)

**Abstract.** We propose a novel method for deconvolving incoherent scatter radar data to recover accurate reconstructions of backscattered powers. The problem is modelled as a hierarchical noise-perturbed deconvolution problem, where the lower hierarchy consists of an adaptive length-scale function that allows for a non-stationary prior and as such enables adaptive recovery of smooth and narrow layers in the profiles. The estimation is done in a Bayesian statistical inversion framework as a two-step procedure, where hyperparameters are first estimated by optimisation and followed by an analytical closed-form solution of the deconvolved signal. The proposed optimisation based method is compared to a fully probabilistic approach using Markov Chain Monte Carlo techniques enabling additional uncertainty quantification. In this paper we examine the potential of the hierarchical deconvolution approach using two different prior models for the length-scale function. We apply the developed methodology to compute the backscattered powers of measured Polar Mesospheric Winter Echoes, as well as Summer Echoes, from the EISCAT VHF radar in Tromsø, Norway. Computational accuracy and performance are tested using a simulated signal corresponding to a typical background ionosphere and a sporadic E layer with known ground-truth. The results suggest that the proposed *hierarchical deconvolution* approach can recover accurate and clean reconstructions of profiles, and the potential to be successfully applied to similar problems.

## 1 Introduction

Radars are devices that emit electromagnetic radiation and detect echo signals scattered from distant targets. In the simplest case, range (distance) resolution of a radar is determined by duration of the transmitted pulses. Since transmission of simple pulses typically produces an insufficient signal-to-noise ratio (SNR), many radar applications are based on long, phase-coded pulses. The coded pulses consist of sequences of short elementary pulses ('bits'), and phase of the carrier frequency signal is changed at the boundaries of the elementary pulses. When an echo signal from a phase-coded pulse is processed with a decoding filter, the filter output resembles an echo of a very powerful elementary pulse. An introduction to different radar systems and pulse compression is given for example by Skolnik (2008).

Incoherent scatter radars (ISR) (Beynon and Williams, 1978) detect radio wave scattering from thermal fluctuations in the partially ionized plasma in the Earth's ionosphere. Very high powers, large antennas, and sophisticated phase-coding techniques

are needed to detect the extremely weak scattering. One problem area of ISR experiment design is selection of the bit length (elementary pulse duration) in phase-coded pulses. The optimal bit length may vary from about a microsecond in certain narrow layers in the lower ionosphere up to a millisecond in the topside ionosphere. The layered phenomena include Polar Mesospheric Summer Echoes (PMSE) (Rapp and Lübken, 2004), Polar Mesospheric Winter Echoes (PMWE) (Kirkwood, 2007), sporadic E layers (Es) (Mathews, 1998), and echoes from hard targets like satellites (Markkanen et al., 2005) and meteors (Pellinen-Wannberg and Wannberg, 1994). The bit length can be optimized for one selected part of the ionosphere only, while the other parts are observed with sub-optimal resolutions.

A common limitation of different decoding filters is their inability to reach resolutions better than the bit length, because this would lead to zeros in Fourier spectra of the decoding filter (Lehtinen et al., 2004). This is an obvious issue in ISR observations, which are typically optimised for resolutions coarser than the typical length scales of the layered phenomena. One way to avoid limitations of the decoding filters is to deconvolve the received signal, for instance by means of statistical inversion. In case of an ISR the deconvolution is typically performed in power domain, that is, one deconvolves range profiles of the signal autocorrelation function instead of the signal itself. Range side-lobes have been suppressed from both zero lags (backscattered power profiles) and from non-zero lags of the autocorrelation function by Huuskonen et al. (1988) and Pollari et al. (1989). Damtie et al. (2002) extended the technique to produce range resolutions considerably finer than the modulation bit length. All these techniques used decoding filters prior to calculation of the lag profiles whenever possible. Techniques that do not include decoding filters have been published by Virtanen et al. (2008) and by Nikoukar et al. (2008).

While deconvolution of ISR signals is possible without regularisation in some cases, most of the published techniques use prior models to suppress noise from the deconvolved lag profiles. Huuskonen et al. (1988) and Pollari et al. (1989) used only boundary conditions to suppress range side-lobes, while Damtie et al. (2002) and Nikoukar et al. (2008) used first-order difference priors. However, the stationary prior model cannot be optimised for the wide range of length scales in the ionosphere. Narrow layers tend to be smoothed out if the prior is tuned for smooth parts of the lag profile, while a prior tuned for the narrow layers leaves the other parts of the profile with an unacceptable noise level. This is also the reason why the option to use priors has been removed from later versions of the technique of Virtanen et al. (2008). Instead, an analysis technique that employs non-stationary priors when fitting plasma parameters to the deconvolved lag profiles was recently published by Virtanen et al. (2021). Also the different "full profile" solvers for the plasma parameter fits (Holt et al., 1992; Lehtinen et al., 1996; Hysell et al., 2008) incorporate prior information in the solutions, but sometimes in a way that is not easy to explain in terms of actual length-scales. In related applications mixed priors have been investigated by Repetti et al. (2014), but still require separate tuning for varying length-scales and represents a common limitation.

Even non-stationary prior models are problematic in practice, because one cannot know the actual length-scale of the iono-sphere at each altitude beforehand. The layered phenomena are sporadic in nature, and altitude and thickness of the layers vary. An automatic system to optimize the length-scales is thus desirable. In this paper we use hierarchical prior models following from Arjas et al. (2020b) in deconvolution of ISR power profiles in presence of strong, narrow layers. Our aim is to study the performance of these prior models in deconvolution of ISR data and to show that they are a potential solution to the problem of variable length-scales, without the need to explicitly control the weighting, that is, the length-scale estimate is

adaptively estimated from the profiles during an efficient optimisation approach. As a relatively simple test case, we consider post-processing of power profiles calculated from decoding filter output, but with oversampled signal to enable resolutions finer than the modulation bit length. The correctness of the recovered profiles is first evaluated with a synthetically generated data corresponding to a typical background ionosphere and a sporadic E layer. We then continue to evaluate the potential of the method for real measurements from the EISCAT VHF radar in Tromsø, Norway. Our main example analyses the performance for Polar Mesospheric Winter Echoes recorded on 24/11/2006. Additionally, we present results for Polar Mesospheric Summer Echoes recorded on 12/08/2018 to showcase the ability of the proposed method for different scenarios. Our results suggest the hierarchical approach can successfully improve the resolution of narrow layers while effectively improving SNR of smooth layers. We expect that the hierarchical deconvolution approach can be effectively utilised in similar deconvolution problems. For easy reproducibility, we will provide accompanying codes with the publication.[1]

This paper is organised as follows: we first discuss the radar decoding techniques and the signal formation task for incoherent scatter radars in Sect. 2. We then introduce the proposed model of *hierarchical deconvolution* in Sect. 3 and its implementation in Sect. 4, either using an efficient optimisation approach or for full uncertainty quantification with Markov chain Monte Carlo (MCMC) methods. The synthetic and real data experiments are then presented and discussed in Sect. 5. A general discussion on the proposed method is presented in Sect. 6 and final conclusions are presented in Sect. 7.

## 2 Radar decoding techniques

The radar transmission signal can be expressed as product of a coherent sinusoidal carrier signal $\mathrm{car}(t)$ and a transmission envelope $\mathrm{env}(t)$, where $t$ is time. The carrier frequency component can be neglected in radar signal processing, because it can be eliminated by means of quadrature sampling (Lehtinen and Huuskonen, 1996). We can thus assume that the transmitted waveform is

$$z_T(t) = \mathrm{env}(t), \tag{1}$$

which is complex-valued in general.

The transmission envelope of a phase-coded pulse can be expressed as convolution of an elementary pulse $v(t)$ and a coding filter $h_c(t)$. The coding filter is given by

$$h_c(t) = \sum_{k=1}^{N} a_k \delta(t - kt_b), \tag{2}$$

where $a_k$ are complex, unit length code coefficients, $\delta(t)$ is the Dirac delta function, and $t_b$ denotes the bit length of the code. The elementary pulse $v(t)$ is typically a boxcar function of length $t_b$, because this choice leads to a constant transmitted power during a pulse. $N$ is the number of bits in the code.

When the transmitted waveform is scattered from a coherent target, the scattered signal entering the radar receiver can be expressed as a convolution of the transmitted waveform $z_T(t)$ and a complex back-scattering coefficient $\sigma(t)$, which absorbs

---

[1]With paper acceptance

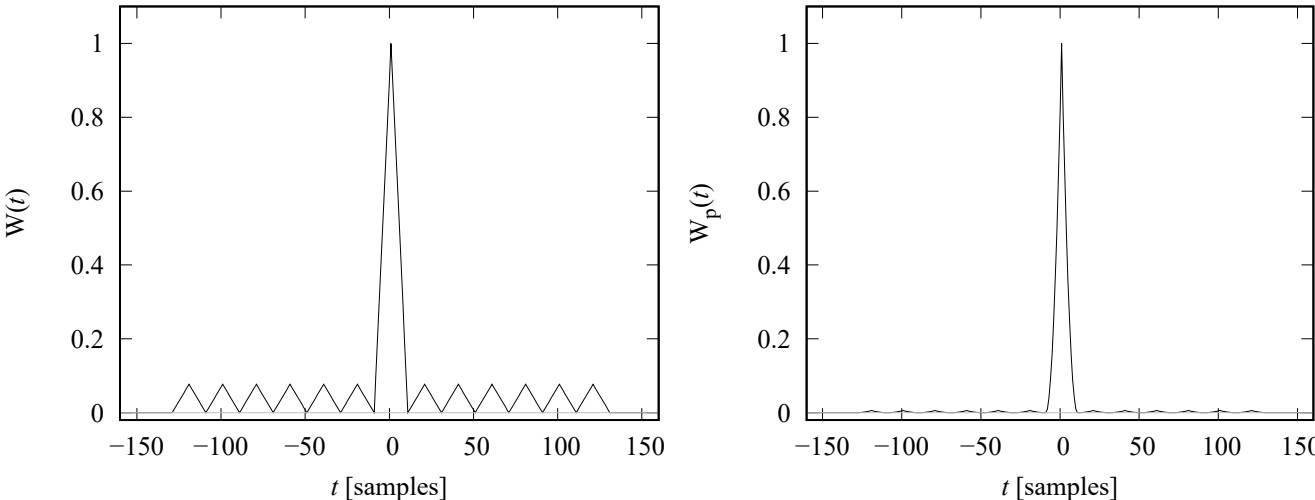

**Figure 1.** Point spread function (left) and zero-lag range ambiguity function (right) of a 13-bit Barker code.

the effects of the radar cross-section of the target, its radial velocity (Doppler shift), and distance. The signal entering the radar receiver is then convolved with the receiver impulse response $p(t)$ and digitised.

The simplest decoding filter is the matched filter, in which the filter impulse response is a time-inverted, complex-conjugated copy of the coding filter,

$$h_m = \overline{h_c}(-t), \tag{3}$$

where the line denotes a complex conjugate. Merging all the convolutions together, we get the matched filter output signal

$$z_m(t) = (v * h_c * \sigma * p * h_m)(t) = (W * \sigma)(t) \tag{4}$$

where the asterisk denotes the convolution operation, $W(t)$ is the point spread function, which is the matched filter output of an echo from a point target, and $\sigma$ is the complex backscattering coefficient.

When the elementary pulse $v(t)$ and the receiver impulse response $p(t)$ are identical the point spread function produced by a matched filter is the autocorrelation function of the transmitted waveform. The matched filter optimises SNR of the decoded signal, but it produces so-called range side-lobes, because the autocorrelation function of a phase-coded pulse with constant amplitude cannot be exactly identical with the autocorrelation function of an elementary pulse. These range side-lobes can be suppressed with inverse filtering with the cost of decreased SNR (for example Damtie et al., 2008, and references therein). If amplitude modulation is available, one can use also the so-called perfect and almost perfect pulse compression codes (Lehtinen et al., 2009; Roininen and Lehtinen, 2013; Roininen et al., 2014a), which do not produce side-lobes in matched filtering. However, while the perfect pulse-compression codes allow one to optimally exploit the energy emitted by the transmitter, the amplitude modulation reduces the total emitted energy.

The incoherent scatter signal is not coherent in time, but scattering from a narrow range interval is a zero-mean random process with finite decorrelation time. Although derived for a coherent target, the above model can be used for decoding individual transmitted pulses of a VHF frequency radar in the D region (below 90 km altitude) and E region (90 - 150 km) of the ionosphere, where decorrelation time of the random scattering process is sufficiently long. Due to its random nature, the backscattered signal itself is usually not of interest, but the power spectral density of the scattering process contains information of properties of the ionospheric plasma. In this paper we consider the backscattered power, which is proportional to the ionospheric electron density to a good approximation in the D and E regions. The backscattered power can be estimated as mean squared amplitude of several subsequent pulses,

$$P_m(t) = \frac{1}{M} \sum_{l=1}^{M} |z_m^l(t)|^2 = \frac{1}{M} \sum_{l=1}^{M} \left| \left( W^l * \sigma^l \right)(t) \right|^2 \approx (P * W_P)(t), \tag{5}$$

where $M$ is the number of averaged pulses, $l$ is the pulse index, $P(t)$ is the true mean backscattered power, and $W_P(t)$ is the range ambiguity function,

$$W_p(t) = \frac{1}{M} \sum_{l=1}^{M} W_P^l(t) = \frac{1}{M} \sum_{l=1}^{M} \left| W^l(t) \right|^2. \tag{6}$$

The range ambiguity functions can be defined also for non-zero lags of the autocorrelation function in general, but the zero-lag version is sufficient for the present work.

The point spread function and range ambiguity function of a commonly used 13-bit Barker code (Barker, 1953) are shown in Fig. 1. Both functions contain a high peak at the centre, and several smaller maxima called side lobes. As discussed above, these side lobes cannot be suppressed from transmitted waveforms with constant amplitude by means of matched filtering and inverse filtering could remove the side lobes with the cost of increased noise. Nevertheless, inverse filtering cannot make the main peak narrower than the autocorrelation function of the elementary pulse. The elementary pulses are 10 samples long in Fig. 1, which means that the total width of the main peak is 20 samples in this example.

In order to move from the continuous time signals to discrete samples, we define a vector of discrete backscattered powers $\mathbf{P} = (P_1, P_2, \ldots, P_H)^\mathsf{T}$, where $H$ is the number of discrete range gates. The unknown backscattered powers $\mathbf{P}$ are true backscattered powers from individual range gates. Discrete samples of the measured average backscattered power can then be expressed as a linear combination of the true backscattered powers $\mathbf{P}$,

$$\mathbf{P}_m = \mathbf{A}\mathbf{P} + \varepsilon, \tag{7}$$

where $\mathbf{P}_m$ are discrete samples of power profiles calculated from the matched filter output, the theory matrix $\mathbf{A}$ contains discrete samples of the range ambiguity function in its rows, and $\varepsilon$ is zero-mean noise. The noise term has contributions from thermal noise in the radar receiver, from galactic radio sources, and from the random fluctuations of the target itself. The noise is assumed to be Gaussian white noise, although the so-called "self-noise" contribution from the target itself may violate this assumption in some cases. The effects of self-noise are discussed in Section 6.1. In order to reach resolutions better than the elementary pulse length, we oversample the signal, i.e. use sampling steps shorter than the elementary pulse length. Equation

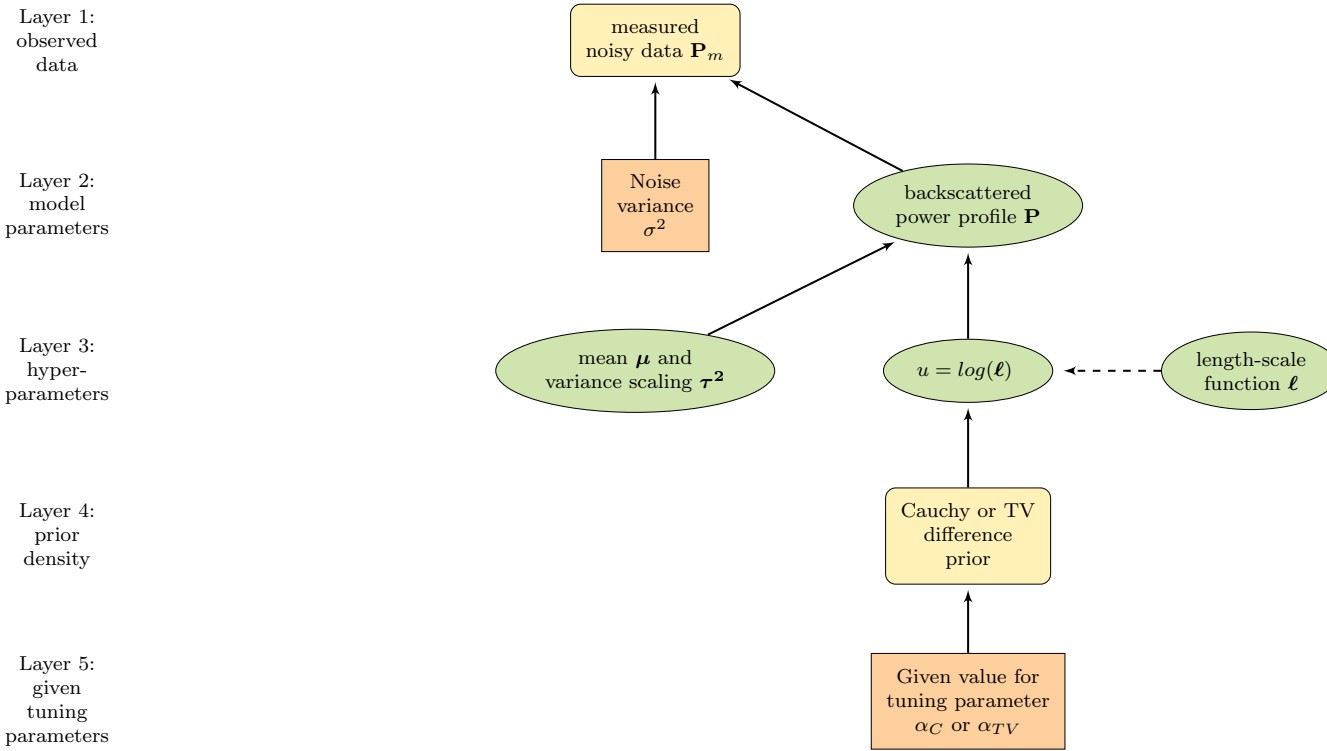

**Figure 2.** The hierarchical structure of the model described as directed acyclic graph. The ovals represent random variables and rectangles given quantities. Solid arrows indicate statistical dependency and the dotted arrow a functional relationship.

(7) could be defined also for signal powers calculated before matched filtering, for non-zero lags of the autocorrelation function, and for a sequence of individual pulses, but this version is sufficient for the present work. For a more detailed and more general derivation, see Virtanen et al. (2008). The true backscattered power profile **P** and its error covariance can be then computed by solving Equation (7) by means of statistical inversion, as will be outlined in the next section.

## 3 Bayesian hierarchical model for decoding

The hierarchical structure and summary of the proposed model is presented in Fig. 2. The aim is to estimate the back-scattered powers **P** in Equation (7), represented by layer 2 in Fig. 2, from the measured data $\mathbf{P}_m$ in layer 1. The recovery problem is modelled as a noise-perturbed deconvolution problem and the estimation is done in a Bayesian statistical inversion framework. For a detailed discussion on Bayesian inversion, we refer to Kaipio and Somersalo (2004). We model the discrete backscattered powers with a locally adaptive Matérn process, allowing us to specify certain smoothness and regularity conditions. The

regularity of the Matérn process is governed by length-scaling $\ell(\cdot)$, represented in layer 3 of Fig. 2, that is also a random process depending on time $t$.

In summary, we are interested in the conditional posterior distribution of $\mathbf{P}$ given the measured data $\mathbf{P}_m$ and discretised length-scaling $\boldsymbol{\ell}$, where the full posterior density of $\mathbf{P}$ and $\boldsymbol{\ell}$ is given by Bayes' formula

$$p(\mathbf{P},\boldsymbol{\ell}|\mathbf{P}_m) = \frac{p(\mathbf{P}_m|\mathbf{P},\boldsymbol{\ell})p(\mathbf{P},\boldsymbol{\ell})}{p(\mathbf{P}_m)} \propto \frac{1}{\sqrt{(2\pi)^n \det(\sigma^2\mathbf{I})}} \exp\left(-\frac{1}{2}(\mathbf{P}_m - \mathbf{AP})^{\mathsf{T}}(\sigma^2\mathbf{I})^{-1}(\mathbf{P}_m - \mathbf{AP})\right) p(\mathbf{P}|\boldsymbol{\ell})p(\boldsymbol{\ell}|\alpha). \tag{8}$$

Here $p(\mathbf{P}_m|\mathbf{P},\boldsymbol{\ell})$ is the likelihood function that depends on the assumed distribution of the noise; $\varepsilon \sim \mathcal{N}(\mathbf{0},\sigma^2\mathbf{I})$ and due to conditional independence is just $p(\mathbf{P}_m|\mathbf{P})$. We factor the joint prior density $p(\mathbf{P},\boldsymbol{\ell})$ hierarchically as $p(\mathbf{P},\boldsymbol{\ell}) = p(\mathbf{P}|\boldsymbol{\ell})p(\boldsymbol{\ell}|\alpha)$, where $p(\mathbf{P}|\boldsymbol{\ell})$ is the prior density concerning assumptions of $\mathbf{P}$ given hyperparameters $\boldsymbol{\ell}$, $p(\boldsymbol{\ell}|\alpha)$ is the prior density for hyperparameters, $\alpha$ is a fixed tuning parameter, and $p(\mathbf{P}_m)$ is a normalising constant, which we can omit in the analysis. The vector $\boldsymbol{\ell}$ consists of hyperparameters that control the properties (discussed in Sect. 3.1) of $\mathbf{P}$ and therefore must be estimated. Following Arjas et al. (2020b), we solve the inverse problem of computing $\mathbf{P}$ in Equation (7) in two parts (described in more detail in Sect. 4):

(i) Find the maximum a posteriori (MAP) estimate for $\boldsymbol{\ell}$, that is $\widehat{\boldsymbol{\ell}} = \arg\max_{\boldsymbol{\ell}} p(\boldsymbol{\ell}|\mathbf{P}_m)$.

(ii) Given $\widehat{\boldsymbol{\ell}}$, recover the conditional posterior distribution of $\mathbf{P}$ with formulas given in closed-form.

## 3.1 Prior model for $\mathbf{P}$

Gaussian processes (GP) are smoothness priors frequently used in machine learning (Rasmussen and Williams, 2006) and inverse problems (Roininen et al., 2014b). In this study, we assume a GP prior for $\mathbf{P}$, denoted by $\mathbf{P} \sim \mathcal{GP}(\mu,\mathbf{C}_\ell)$. GPs require the selection of a mean function and covariance function. We assume the mean function $\mu$ to be zero, which is a simplified assumption, but sufficient to detect the desired non-stationary features in this study. The covariance function $C(\cdot,\cdot)$, on the other hand, assigns a covariance between two points $t$, $t' \in \mathbb{R}$ based on their distance and thereby regulates the properties of the process. A usual choice is the Matérn covariance function, defined as

$$C(t,t') = \tau^2 \frac{2^{1-\nu}}{\Gamma(\nu)} \left(\frac{|t-t'|}{\ell}\right)^\nu K_\nu\left(\frac{|t-t'|}{\ell}\right), \tag{9}$$

where $\tau^2$ is the magnitude (variance scaling), $\nu$ the smoothness, $\ell$ the length-scale, and $K_\nu$ the modified Bessel function of the second kind of order $\nu$. The covariance function is stationary and lacks adaptability in the sense of non-stationarity.

Matérn fields can be also defined through stochastic differential equations (SDE) (Lindgren et al., 2011; Roininen et al., 2014b). We choose $\nu = p + 0.5$, $p \in \mathbb{N}$, as this provides a Markov approximation for the model. By the construction, the square-root of the inverted covariance matrix has a tridiagonal structure – which is numerically convenient. Fixing $\nu = 1.5$ and $\tau^2 = 1$, a GP with Matérn covariance can be defined as

$$(1 - \ell^2\Delta)P = 2\sqrt{\ell}w, \tag{10}$$

where $\Delta$ is the Laplace operator and $w \sim \mathcal{N}(0,1)$. To increase the flexibility of the GP, one can introduce spatially/temporally varying length-scale depending on $t$ (Roininen et al., 2019), and the SDE becomes

$$(1 - \ell(t)^2 \Delta)P = 2\sqrt{\ell(t)}w. \tag{11}$$

The continuous model (11) must be discretised for computations. After discretisation we denote $\mathbf{C}_\ell = (\mathbf{L}_\ell^\mathsf{T}\mathbf{L}_\ell)^{-1}$ and obtain the matrix-vector representation $\mathbf{L}_\ell\mathbf{P} = \mathbf{w}$, where the multipliers of $\mathbf{w}$ are absorbed into $\mathbf{L}_\ell$. This gives the prior distribution of $\mathbf{P}$ as $\mathbf{P} \sim \mathcal{GP}(0, (\mathbf{L}_\ell^\mathsf{T}\mathbf{L}_\ell)^{-1})$. By design, $\mathbf{L}_\ell$ is a tridiagonal matrix (Roininen et al., 2014b), motivated by the Markov approximation that leads to a computationally useful presentation. Moreover, this also allows us to model $\ell$ via increments, thus simplifying both the model and computations. The explicit representation of $\mathbf{L}_\ell$ is given in the Section 4.

### 3.2 Prior model for $\ell$

The length-scale function is estimated from the data itself. It is required to be positive so we make use of a logarithmic transformation $u(t) = \log(\ell(t))$ and estimate $u(\cdot)$ instead of $\ell(\cdot)$, see layer 3 in Fig. 2. In the following we consider two different hyperprior models: a Cauchy difference prior and total variation (TV) prior (Laplace difference prior). The probability density functions of Cauchy and Laplace distributions are given as $p(x) \propto ((x - x_0)^2 + s^2)^{-1}$ and $p(x) \propto \exp(|x - x_0|s^{-1})$, 190 respectively, where $x_0$ is the center and $s$ is the scaling of the distribution. Essentially we assume that

$$u(t_i) - u(t_{i-1}) \overset{\text{i.i.d.}}{\sim} \text{Cauchy}(0, \alpha_\text{C}), \text{ or}$$
$$u(t_i) - u(t_{i-1}) \overset{\text{i.i.d.}}{\sim} \text{Laplace}(0, \alpha_\text{TV}), \tag{12}$$

where $\alpha_\text{C}$ or $\alpha_\text{TV}$ act as a tuning parameter, represented as layer 4 in Fig. 2, controlling the level of regularity and must be chosen for each dataset individually. The priors then adaptively regularise the length-scale function and stabilise the posterior distribution. Statistically both Cauchy and Laplace distributions are long-tailed distributions which favor having near zero 195 differences in consecutive $u$-values but still allow some non-zero differences (jumps). Generally, the number of non-zero differences is dependent on the value of the tuning parameter.

## 4 Signal recovery via optimisation and MCMC schemes

We can now solve the inverse problem in Equation (7) to obtain the backscattered powers $\mathbf{P}$ with the tools presented above. First we need to compute the MAP estimate of $\mathbf{u}$, denoted as $\hat{\mathbf{u}}$, which is found by maximising the logarithm of the marginal 200 posterior density. This could be done for instance by available optimisation algorithms, such as the limited memory BFGS algorithm (Liu and Nocedal, 1989) implemented in the R function *optim()*, which computes the derivatives automatically by finite differences. This approach was taken in Arjas et al. (2020b), but creates a computational bottleneck. In this study, to improve reconstruction times we specify the gradient of the target function explicitly for a more efficient computation, instead of time consuming finite-difference approximations of the gradients. First, we note that the logarithm of the marginal posterior

density of $\mathbf{u}$ is given as

$$\log p(\mathbf{u}|\mathbf{P}_m) = -\frac{1}{2}\log\det\mathbf{K} - \frac{1}{2}\mathbf{P}_m^{\mathsf{T}}\mathbf{K}^{-1}\mathbf{P}_m - b_{\mathrm{C/TV}},$$

$$b_{\mathrm{C}} = \sum_{i=2}^{n}\log((u_i - u_{i-1})^2 + \alpha_{\mathrm{C}}^2),$$

$$b_{\mathrm{TV}} = \alpha_{\mathrm{TV}}^{-1}\sum_{i=2}^{n}\sqrt{(u_i - u_{i-1})^2 + \psi}, \tag{13}$$

where $b_{\mathrm{C/TV}}$ is either the log Cauchy or TV prior density given as above, $\mathbf{K} = \mathbf{A}\mathbf{C}_\ell\mathbf{A}^{\mathsf{T}} + \sigma^2\mathbf{I}$, and $\psi = 10^{-8}$ is a small parameter to make $b_{\mathrm{TV}}$ differentiable. The $i$th entry of the gradient of (13) can then be computed as

$$(\nabla\log p(\mathbf{u}|\mathbf{P}_m))_i = -\frac{1}{2}\mathrm{tr}\left(\mathbf{K}^{-1}\frac{\partial\mathbf{K}}{\partial u_i}\right) + \frac{1}{2}\mathbf{P}_m^{\mathsf{T}}\mathbf{K}^{-1}\frac{\partial\mathbf{K}}{\partial u_i}\mathbf{K}^{-1}\mathbf{P}_m - \frac{\partial b_{\mathrm{C/TV}}}{\partial u_i},$$

$$\frac{\partial b_{\mathrm{C}}}{\partial u_i} = \frac{2(u_{i+1} - u_i)}{(u_{i+1} - u_i)^2 + \alpha_{\mathrm{C}}^2} - \frac{2(u_i - u_{i-1})}{(u_i - u_{i-1})^2 + \alpha_{\mathrm{C}}^2},$$

$$\frac{\partial b_{\mathrm{TV}}}{\partial u_i} = \frac{2(u_{i+1} - u_i)}{\alpha_{\mathrm{TV}}\sqrt{(u_{i+1} - u_i)^2 + \psi}} - \frac{2(u_i - u_{i-1})}{\alpha_{\mathrm{TV}}\sqrt{(u_i - u_{i-1})^2 + \psi}}. \tag{14}$$

The employed optimisation algorithm converges if it is unable to reduce the relative value of the target function by $10^{-8}$. After finding $\widehat{\mathbf{u}}$, the estimate for the length-scale function is simply given as $\widehat{\ell} = \exp(\widehat{\mathbf{u}})$, which is then used to construct the tridiagonal matrix $\mathbf{L}_{\widehat{\ell}}$. The diagonal entries of the matrix at row $i$ are given as $b_i = (1 + 2\widehat{\ell}_i^2)/\sqrt{4\widehat{\ell}_i}$ and entries left and right of the diagonal as $a_i = -\widehat{\ell}_i^2/\sqrt{4\widehat{\ell}_i}$. We note that the conditional posterior distribution of $\mathbf{P}$ given $\widehat{\ell}$ is Gaussian with mean and covariance given as closed-form solutions. That is, the mean, representing the main quantity of interest, can be computed by

$$\widehat{\mathbf{P}} = \mathbf{C}_{\widehat{\ell}}\mathbf{A}^{\mathsf{T}}(\mathbf{A}\mathbf{C}_{\widehat{\ell}}\mathbf{A}^{\mathsf{T}} + \sigma^2\mathbf{I})^{-1}\mathbf{P}_m, \tag{15}$$

where $\mathbf{C}_{\widehat{\ell}} = (\mathbf{L}_{\widehat{\ell}}^{\mathsf{T}}\mathbf{L}_{\widehat{\ell}})^{-1}$. In the following, we consider the estimate $\widehat{\mathbf{P}}$ as the final recovered signal.

Alternatively to the above presented MAP estimate, inference for Equation (8) can be performed using Markov Chain Monte Carlo (MCMC) techniques. Instead of finding the posterior mode, the idea of MCMC is to draw a sample from the posterior distribution which can be used to estimate posterior expectations and variances for uncertainty quantification. For discussion of MCMC methods, see Robert and Casella (2009). In this study, a combination of Gibbs sampling and Metropolis–Hastings algorithm is used. The same sampling scheme has previously been utilised in Roininen et al. (2019). The idea is to alternatingly sample $\mathbf{P}$ from its (Gaussian) conditional distribution given $\mathbf{u}$, and each component $u_i$ individually from their conditional distributions given $\mathbf{u}_{-i}$ and $\mathbf{P}$. The distribution of $u_i|\mathbf{u}_{-i},\mathbf{P}$ is not of a known form, so Metropolis–Hastings algorithm must be utilised. After suitably many MCMC iterations, a sample from the joint posterior distribution of $(\mathbf{P}, \mathbf{u})$ is received. This has the advantage of providing full uncertainty quantification, but due to sampling is computationally much more expensive.

### 4.1 Model overview

To conclude the methodological section we summarise the full hierarchical model here and provide pseudocode for computing the reconstructions. The full model as depicted in Fig. 2 can be written in a hierarchical form as

$$\mathbf{P}_m|\mathbf{P} \sim \mathcal{N}(\mathbf{AP}, \sigma^2\mathbf{I}),$$
$$\mathbf{P}|\boldsymbol{\ell} \sim \mathcal{GP}(\mathbf{0}, \mathbf{C}_\ell), \quad \mathbf{u} = \log(\boldsymbol{\ell})$$
$$u(t_i) - u(t_{i-1}) \sim \mathrm{Cauchy/Laplace}(0, \alpha_{\mathrm{C/TV}}) \tag{16}$$

The primary algorithm for recovering a point estimate by computing the MAP estimator is presented as pseudocode in Algorithm 1. Alternatively, when computing the full posterior distribution, one needs to perform MCMC inference. The corresponding MCMC scheme used here is summarised in Algorithm 2 2.

---

**Algorithm 1** MAP inference

---

1: $\widehat{\mathbf{u}} = \arg\max_{\mathbf{u}}(\log p(\mathbf{u}|\mathbf{P}_m))$ (Eq.13)
2: $\widehat{\boldsymbol{\ell}} = \exp(\widehat{\mathbf{u}})$
3: Construct tridiagonal matrix $\mathbf{L}_{\widehat{\ell}}$
4: $\mathbf{C}_{\widehat{\ell}} = (\mathbf{L}_{\widehat{\ell}}^{\mathsf{T}} \mathbf{L}_{\widehat{\ell}})^{-1}$
5: $\widehat{\mathbf{P}} = \mathbf{C}_{\widehat{\ell}} \mathbf{A}^{\mathsf{T}} (\mathbf{A}\mathbf{C}_{\widehat{\ell}}\mathbf{A}^{\mathsf{T}} + \sigma^2\mathbf{I})^{-1}\mathbf{P}_m$

---

## 5 Synthetic test case and EISCAT experiment analysis

We test performance of the proposed *hierarchical deconvolution* using a synthetic radar signal and real measurements from the EISCAT VHF radar in Tromsø, Norway. The data are voltage level quadrature samples of the radar echo signal and the transmitted waveform. The data are decoded by means of matched filtering and average profiles of backscattered power are calculated from the decoded data. Range ambiguity functions are calculated from samples of the transmitted waveforms using the same procedure. The power profiles are then deconvolved to range resolution finer than the modulation bit length using the hierarchical deconvolution. Range side lobes produced by the matched filter are also suppressed in this process. In the deconvolution process one must select a noise variance $\sigma^2$ and a tuning parameter $\alpha$, after which the prior length-scales $\ell_i$ and the reconstructed signal $\hat{\mathbf{P}}$ are solved from the data. To understand the capabilities of the method, we present results for varying parameters of $\alpha$ for both prior choices discussed in Sect. 3.1.

### 5.1 Synthetic test case

To test the accuracy of the method and obtained results, a simulated signal corresponding to a typical background ionosphere and a sporadic E layer was created. The background electron density profile was taken from the International Reference Ionosphere (Bilitza et al., 2017) on summer daytime conditions, and a narrow layer with peak electron density $N_e = 10^{12}\ \mathrm{m}^{-3}$ was

**Algorithm 2** MCMC inference

**Require:** Initial parameter values $\mathbf{P}^0, \mathbf{u}^0, \boldsymbol{\ell}^0 = \exp(\mathbf{u}^0), \widehat{\mathbf{P}} = \mathbf{P}^0$

1: $t \leftarrow 0$
2: $T \leftarrow 10^6$
3: **while** $t < T$ **do**
4:      $t \leftarrow t + 1$
5:      $\mathbf{P}^t = \begin{pmatrix} \sigma^{-1}\mathbf{A} \\ \mathbf{L}_{\ell^{t-1}} \end{pmatrix}^{\dagger} \left( \begin{pmatrix} \sigma^{-1}\mathbf{P}_m \\ \mathbf{0} \end{pmatrix} + \boldsymbol{\delta} \right)$   $\{\boldsymbol{\delta} \sim \mathcal{N}(\mathbf{0}, \mathbf{I}), \dagger : \text{pseudoinverse}\}$
6:      $\boldsymbol{\ell}^* = \boldsymbol{\ell}^{t-1}$
7:      **for** j = 1:n **do**
8:          Propose $u_j \sim \mathcal{N}(u_j^{t-1}, s_j)$
9:          $\ell_j^* = \exp(u_j)$
10:          $\alpha = \log\det(\mathbf{L}_{\ell^*}) - \frac{1}{2}\mathbf{P}^{t\mathsf{T}}\mathbf{L}_{\ell^*}^{\mathsf{T}}\mathbf{L}_{\ell^*}\mathbf{P}^t - \log\det(\mathbf{L}_{\ell^{t-1}}) + \frac{1}{2}\mathbf{P}^{t\mathsf{T}}\mathbf{L}_{\ell^{t-1}}^{\mathsf{T}}\mathbf{L}_{\ell^{t-1}}\mathbf{P}^t - b_{\text{C/TV}}^* + b_{\text{C/TV}}^{t-1}$
11:          Draw $\beta \sim \mathrm{U}(0,1)$
12:          **if** $\log(\beta) < \alpha$ **then**
13:             $\ell_j^t = \ell_j^*$
14:          **else**
15:             $\ell_j^* = \ell_j^{t-1}$
16:          **end if**
17:      **end for**
18:      $\mathbf{u}^t = \log(\boldsymbol{\ell}^*)$
19:      $\widehat{\mathbf{P}} = \frac{t-1}{t}\widehat{\mathbf{P}} + \frac{1}{t}\mathbf{P}^t$
20: **end while**

added on the background profile at 105 km altitude. Random noise with power spectral density corresponding the incoherent scatter process with the modelled plasma parameters was then generated at discrete altitudes, and the altitude profiles were convolved with the selected transmitted waveform. Finally, uncorrelated random noise was added to the signal to represent the background noise in measurements. The same simulation software was used in Virtanen et al. (2009) and the procedure is very similar with that used by Swoboda et al. (2017). The simulation was carried out assuming 1 MHz sampling rate and Barker coded pulses with the 13-bit code and 10 $\mu$s bit length, which produces 10 samples from each bit of the code. The simulated data were decoded by means of matched filtering and average power profiles were calculated over 665 subsequent transmitted pulses, which leads to 1 s time resolution. The 10 $\mu$s bit length produces 1.5 km nominal range resolution in matched filter decoding, while the 1 MHz sampling rate enables range resolutions down to 150 m in the deconvolution process.

An important aspect in the reconstruction is the choice of tuning parameter $\alpha$ for the priors in (12), which will influence the reconstructed characteristics. To find such an optimal tuning parameter $\alpha$, we performed a grid search for each signal and prior, to find the parameter such that the mean squared error between the estimate of backscattered power signal $\boldsymbol{P}$ and ground-

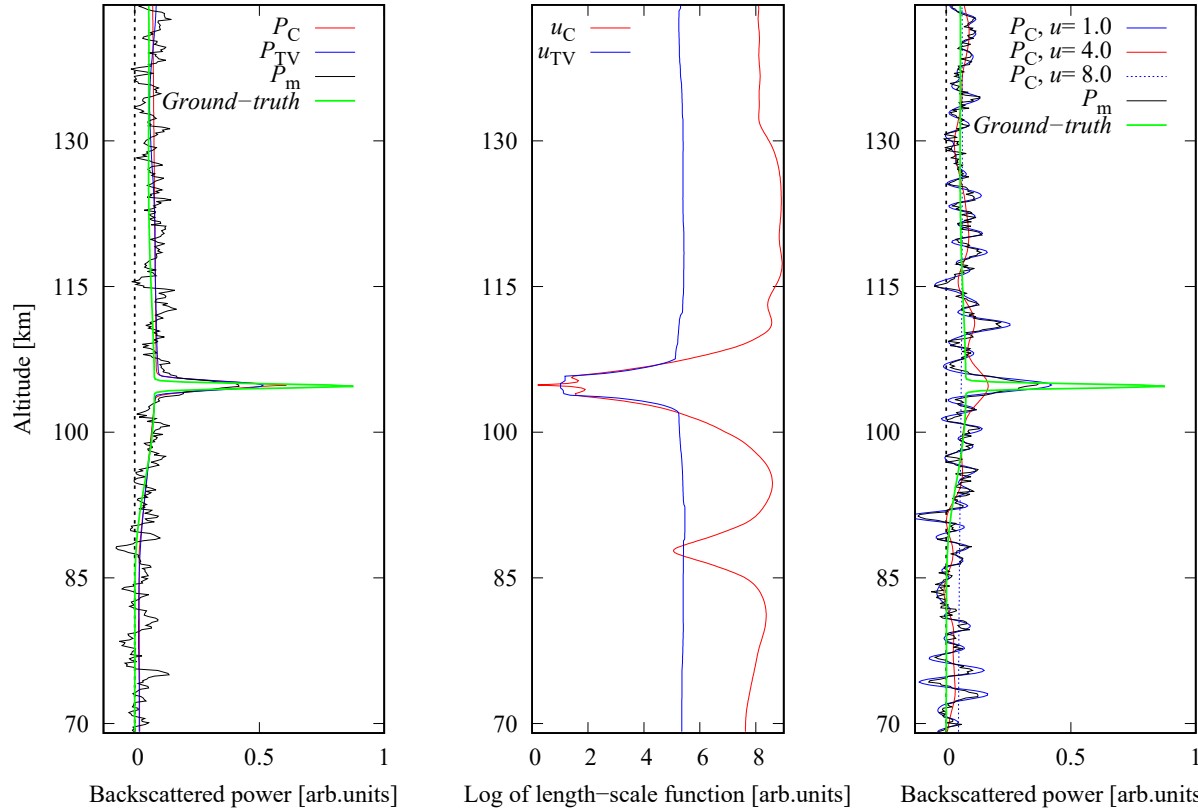

**Figure 3.** Synthetic power profile reconstruction and true power profile (green line). Left panel represents the comparison of Cauchy difference prior (red line) with tuning parameter $\alpha_C = 3.0$ and TV (blue line) prior, with $\alpha_{TV} = 5.0$. Corresponding length-scale functions in the middle. Right panel represents the reconstruction for the Cauchy difference prior for the fixed values of the length-scale function

truth was minimised. The optimal values of the tuning parameters obtained are $\alpha_C = 3.0$ for the Cauchy difference prior, and
$\alpha_{TV} = 5.0$ for the TV prior, were then used for the analysis of the synthetic signal. The presented modelling proves to be robust in recovering the peak power. Specifically, choosing different tuning parameters to estimate the narrow feature with the highest peak power did not significantly improve the quality of the reconstruction in comparison to the optimal values of $\alpha_C$ and $\alpha_{TV}$. The reconstruction of the synthetic power profile, obtained following the procedure outlined in Sect. 4, is presented in Fig. 3. The left panel shows the known true power profile (green line), the average power profile after matched filter decoding (black
line), reconstruction with the Cauchy difference prior (red line), and reconstruction with the TV prior (blue line). The figure demonstrates how the narrow sporadic E layer at 105 km altitude is spread in altitude and how its peak power is decreased in the matched filter decoding. The reconstructions with both Cauchy and TV priors produce layers whose peak power and width are closer to the ground-truth, although neither of them can exactly reproduce the very narrow peak of the layer. The

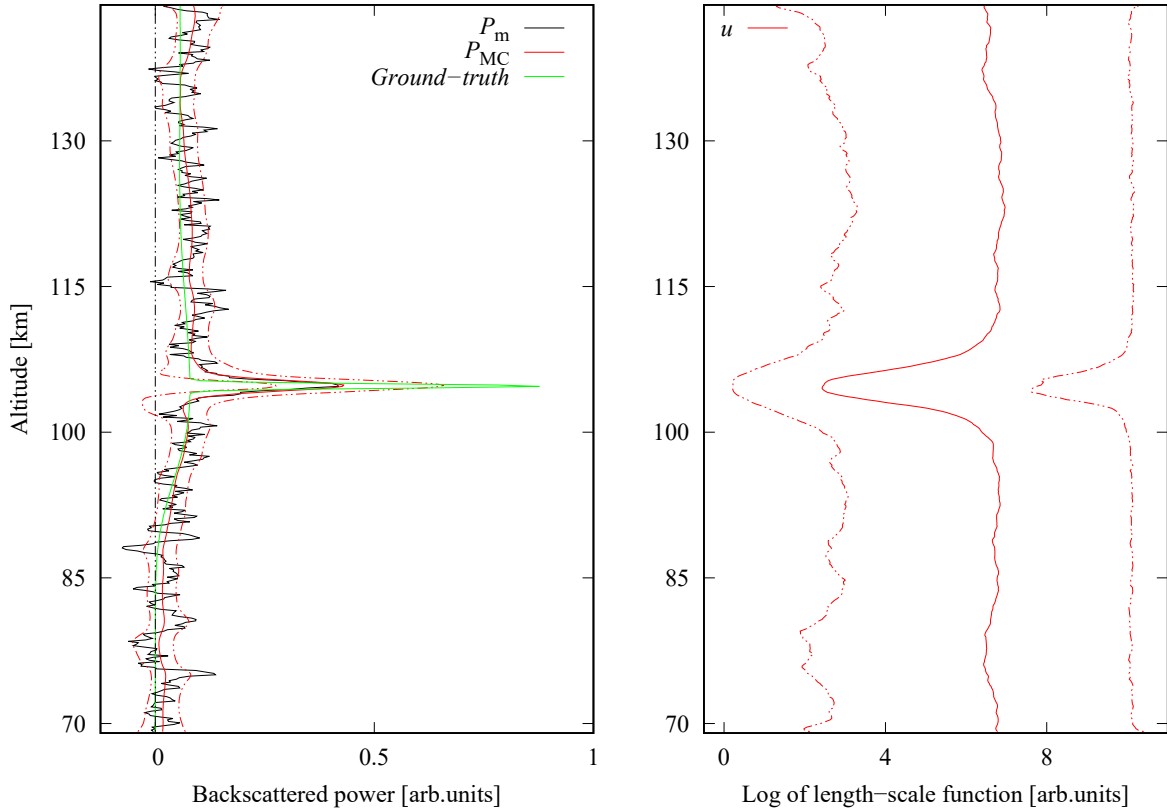

**Figure 4.** Left panel shows synthetic signal reconstruction with MCMC drawn with blue solid line and 95% credible intervals with dashed lines for $\alpha_{\mathrm{MC}} = 0.5$. Reconstruction with the MCMC are plotted in comparison to the true power profile (green line), reconstruction with the Cauchy difference prior (red line). Right panel represents corresponding values of length-scale function in logarithmic scale.

peak power is slightly closer to the ground-truth in the Cauchy difference prior reconstruction. The results depend also on the standard deviation of the measurement error, which was set to $\sigma = 0.1$ (in the arbitrary units used in Fig. 3). The value is larger than standard deviation of the background noise in the averaged data (0.04) to accommodate for the self-noise from the strong scattering layer.

The most important feature in the reconstructions is that while the Es layer was closer to ground-truth in both reconstructions than in the profile calculated from the matched filter outputs, also the smooth parts below and above the Es layer are very close to the ground-truth, with a slight over-estimation, and the SNR is strongly improved from the matched filter output at the same time. This is possible because the prior length-scale is height-dependent and it was estimated as part of the deconvolution process. This is the key difference to a somewhat similar deconvolution by Damtie et al. (2002), who had to select variances of a difference prior at each altitude around the Es layer separately to avoid very noisy results below and above the layer. The

middle panel of Fig. 3 shows the corresponding length-scale functions for the two priors. The length-scales are short at and around the Es layer altitude (105 km) where steep gradients exist, and by factor 3-5 larger in smooth parts of the profile. This illustrated the capability to adapt to the local regularity of the signals to be recovered.

The need for height-dependent length-scale in practice is demonstrated in the third panel of Fig. 3, which shows reconstructions with constant length-scales $u = 1.0$ (blue curve), $u = 4.0$ (red), and $u = 8.0$ (dotted). The Es layer is closer to the ground-truth than the matched filter output (black line) only with the shortest length-scale ($u = 1.0$), which produces a very noisy result in all other parts of the profile. Increasing the length-scale to ($u = 4.0$) practically smooths out the layer from the reconstruction, but the other parts of the profile still suffer from significant noise fluctuations. In addition, altitudes below and above the narrow layer are biased when backscattered power from the layer is spread to adjacent altitudes in the smoothing process. The longest length-scale ($u = 8.0$), which corresponds to the longest length-scales produced by the Cauchy difference prior, practically smoothed out all details and the Es layer when applied to the whole profile. This clearly demonstrates that one cannot select one length-scale that would produce acceptable results in all parts of the profile.

Figure 4 shows reconstruction of the synthetic signal with Cauchy difference prior performed with MCMC estimation. The MCMC specific regularisation parameter was chosen as $\alpha_{\mathrm{MC}} = 0.5$ after visual inspection. The MCMC algorithm was run for 1 000 000 iterations where the burn-in period was one third of the iterations, i.e., the first 333 333 MCMC samples were discarded. The remaining samples were thinned by choosing every 10th realisation to save storage capacity and to reduce the autocorrelation of the Markov chains. The estimate $\widehat{\mathbf{P}}$ was then recovered as the elementwise mean of the Markov chains. Moreover, from MCMC samples, 95% credible intervals were computed by finding the 2.5% and 97.5% quantiles of the estimated posterior distributions. The convergence was examined by the effective sample sizes (ESS) of the Markov chains. The mean ESS was 13500 for the components of $\mathbf{P}$ and 1000 for the components of $\mathbf{u}$. To reach convergence, the computation took about 3 days.

The obtained MCMC reconstruction shows similarities with the MAP reconstruction, while the peak is smoothed out a bit more in the mean and the length-scale function is slightly flatter. However, the credible interval shows the highest uncertainty at the Es layer with a higher variability to larger powers. This additional information on the variability of the reconstructed signal could be used in a further analysis to evaluate the correctness of obtained MAP reconstructions and the range of possible true values.

## 5.2   Polar Mesospheric Winter Echoes

Incoherent scatter radar observations of the D region ionosphere sometimes contain strong, coherent echoes from narrow layers at 50-80 km altitudes. These Polar Mesospheric Winter Echoes (PMWE) are significantly stronger than incoherent scatter echoes from the surrounding ionospheric plasma. We use EISCAT VHF radar data that contain PMWEs to demonstrate the performance of the hierarchical priors for real data. The data are from 24/11/2006, with the start time of the recording at 09:07:39 UT, when the radar was transmitting a special modulation that consists of two 10-bit phase-coded pulses with 12 $\mu$s bit length. The data were recorded with 1 MHz sampling rate, which produced 12 samples from each bit of the code and allows 150 m altitude resolution in theory. The data were decoded with a matched filter that produces a nominal resolution of 1.8 km,

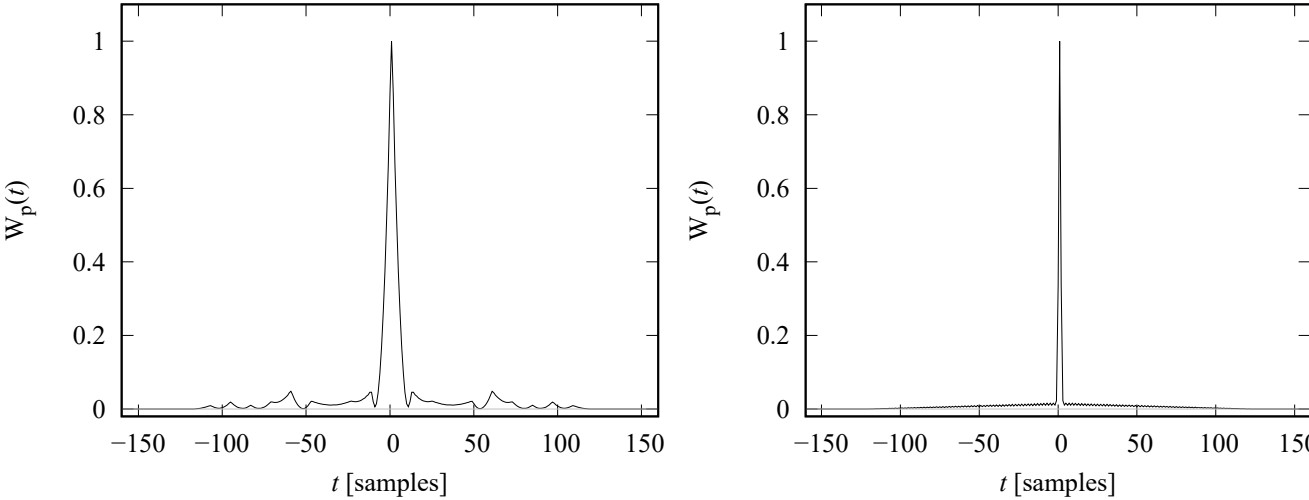

**Figure 5.** Zero-lag range ambiguity functions in the PMWE observation (left) and in the PMSE observation (right). The pair of 10-bit codes used in the PMWE observation produces significant side lobes in matched filter decoding, while the sequence of 128 61-bit codes used in the PMSE observation leads to a smooth pattern of smaller side lobes.

and average profiles of the backscattered power were calculated with 0.9 s (665 pulses) time resolution. The range ambiguity function of the averaged power profiles is shown in the left panel of Fig. 5. Considerable side lobes are produced in matched filtering of the short codes that are not Barker codes.

Computational results are illustrated in Fig. 6 for the Cauchy difference prior with tuning parameter value $\alpha_{\mathrm{C}} = 2.5$ and the TV prior, with $\alpha_{\mathrm{TV}} = 1.0$, respectively. Standard deviation of the measurement error was set to $\sigma = 0.1$ (in the units used in Fig. 6). The value is an order of magnitude larger than the thermal background noise in the time-averaged data (0.01) to accommodate for the significant self-noise from the strong layer. The values of the tuning parameters were carefully chosen

individually to optimise the performance of both reconstructions. The ground truth is not available for the PMWE and PMSE measured signals. As such, the tuning parameters in this case were chosen empirically and were validated visually by professional judgment to improve reconstruction characteristics. Specifically, concentrating on the resolution of the narrow layer and continuity over time (in Figure 7), while maintaining smooth characteristics of the outer layers. The left part of the figure shows the power profile calculated from the matched filter output (black), and reconstructions with the Cauchy (red) and TV (blue)

priors. The right part of the figure illustrates the estimates of the corresponding length-scale functions. Similar to the synthetic signal reconstruction, both priors perform well outside the strong PMWE layer. Around the 65 km peak altitude of the PMWE layer both reconstructions produce higher backscattered powers than in the power profile calculated from the matched filter output. This is expected, because the matched filter tends to smooth out structures narrower than the modulation bit length. While the TV prior produces higher peak power than the Cauchy difference prior, it also seems to slightly spread out the layer

at its upper and lower edges, likely caused by the slight oscillation of the length-scale function, as seen in the right panel of Fig. 6.

The full time-series of backscattered power profiles $\mathbf{P}$ for the Cauchy difference prior and TV prior are shown in Fig. 7. From left to right, the panels are the backscattered power after matched filtering, the reconstruction with the Cauchy difference prior, and the reconstruction with the TV prior. Reconstructions were calculated using the tuning parameter values as $\alpha_\mathrm{C} = 2.5$

and $\alpha_\mathrm{TV} = 1.0$ fixed over the whole time-series, with 5 s time resolution and 150 m range resolution. One should notice that the profiles are not corrected with the signal attenuation by distance squared, and the powers are thus not proportional to the electron density. We present the profiles in this format, because the inversion is performed for uncorrected profiles. Both reconstructions are continuous in time, demonstrating stability of the inversion process. The normal D region echoes above 80 km altitude are very similar with both priors by visual inspection. Maximum amplitude of the PMWE layer is higher and

the peak of the layer is narrower with the TV prior compared to the Cauchy difference prior, but both priors clearly improve resolution of the PMWE layer compared to matched filtering. Most notably, some weak echoes have been smoothed out in TV prior result, whereas the Cauchy difference prior keeps these weaker echoes intact.

### 5.3 Polar Mesospheric Summer Echoes

Polar Mesospheric Summer Echoes (PMSE) are strong, coherent radar echoes from a narrow layer at 80-90 km altitude. Like

PMWEs, also PMSEs are much stronger than the incoherent scatter echoes from the surrounding altitudes. In order to test the flexibility of the proposed *hierarchical deconvolution* framework we additionally analyse EISCAT VHF radar data that contains PMSE echoes, with the start of the recording at 12/08/2018 at 00:00 UT. The radar modulation consisted of a sequence of 128 phase-coded pulses with 2.4 $\mu$s bit length. The backscattered signal was sampled with 1.2 $\mu$s sampling steps, which produces two samples per bit of the code and allows 180 m resolution in the inversion. The corresponding range ambiguity function

of the averaged power profiles is shown on the right panel of Fig. 5. The range side lobes are smooth because the long codes behave reasonably well in matched filtering, and side lobe patterns of each code in the long code sequence are different. In order to assess the capability to reconstruct functions of varying shape, we chose the signal recorded at 00:16:40 UT shown in Fig. 8. The profile is an average over 128 subsequent pulses (0.2 s in time). We compare again Cauchy (with tuning parameter $\alpha_\mathrm{C} = 2.5$) and TV ($\alpha_\mathrm{TV} = 5.0$) priors in the reconstruction of the unknown backscattered power signal shown in the left plot

and the corresponding length-scale on the right part of the figure. Measurement error standard deviation was set to $\sigma = 0.06$ (in the units used in Fig. 8). The value is again considerably larger than the thermal background level (0.005) to accommodate from the self-noise from the very strong layer.

In the selected profile the PMSE has a double-peak structure with a narrow gap between the two peaks. In all others parts except in between the two peaks both priors produce almost identical results, with peak powers considerably larger than the

matched filter output. However, in the valley between the two peaks the TV prior produces some oscillations that do not seem realistic. In particular, the reconstructed power is clearly below the background power as observed above and below the double-peaked PMSE. In contrast, the Cauchy difference prior produces a more realistic smooth valley with minimum power equal to the background power.

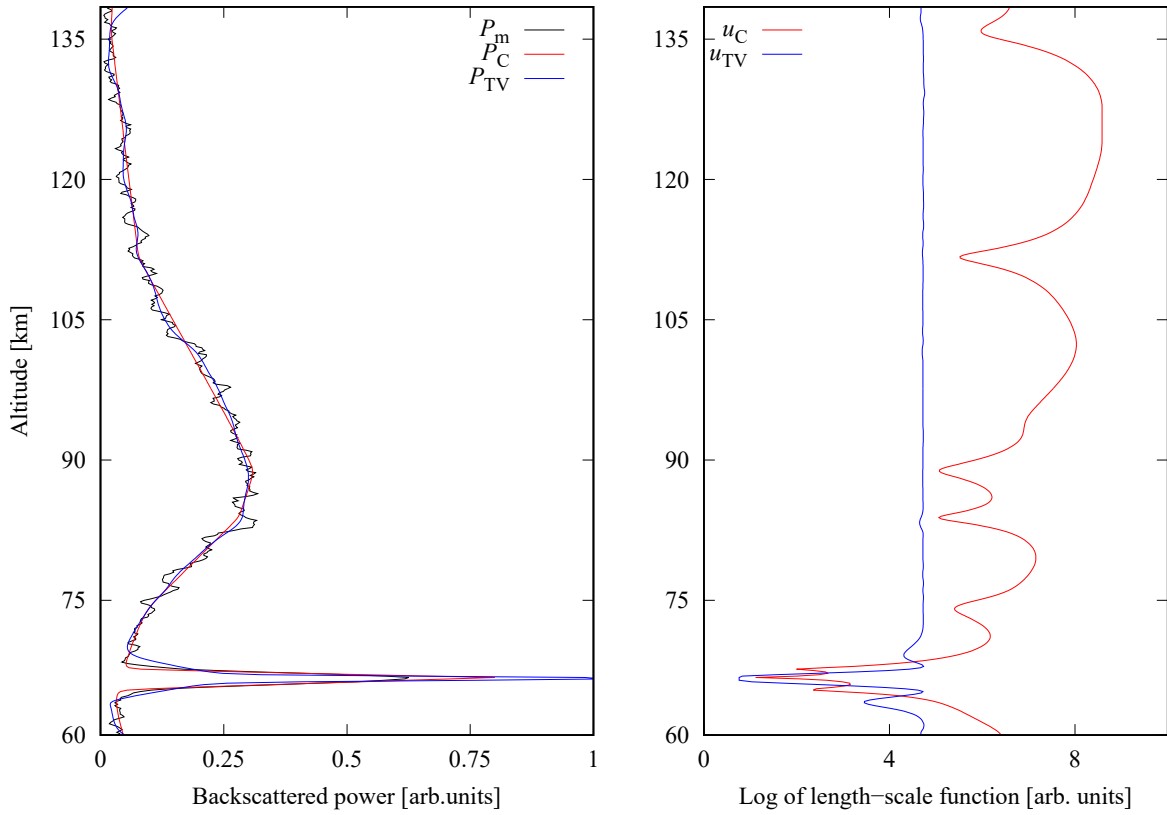

**Figure 6.** PMWE EISCAT data from 24/11/2006, at exact UT time 09:20:48. Comparison of the reconstructions of the unknown signal $P$ with the Cauchy difference prior (red line, $\alpha_C = 2.5$) and TV (blue line, $\alpha_{TV} = 1.0$) prior, on the left. The backscattered power profile after matched filtering $P_m$ are plotted with black line. Right panel shows corresponding values of the length-scale functions in logarithmic scale.

## 6   Discussion

The presented results demonstrate the capability of the proposed *hierarchical deconvolution*. In particular, both priors, Cauchy difference and TV, produce very good results below and above the Es, PMWE, and PMSE layers. The considered hierarchical priors would thus be a useful tool in studies of the smooth background ionosphere in presence of strong, narrow layers, because they can efficiently improve resolution and SNR by suppressing noise fluctuations from smooth parts of the profiles without spreading power from the narrow layers to other parts of the profiles. In addition to the power profile analysis performed in this

paper, the priors could be used in full profile incoherent scatter analysis, in which spatial gradients in plasma parameter altitude profiles are controlled with prior models. In particular, the Bayesian filtering module (Virtanen et al., 2021) could benefit from priors that would automatically adapt to sporadic E layers and other strong, narrow targets.

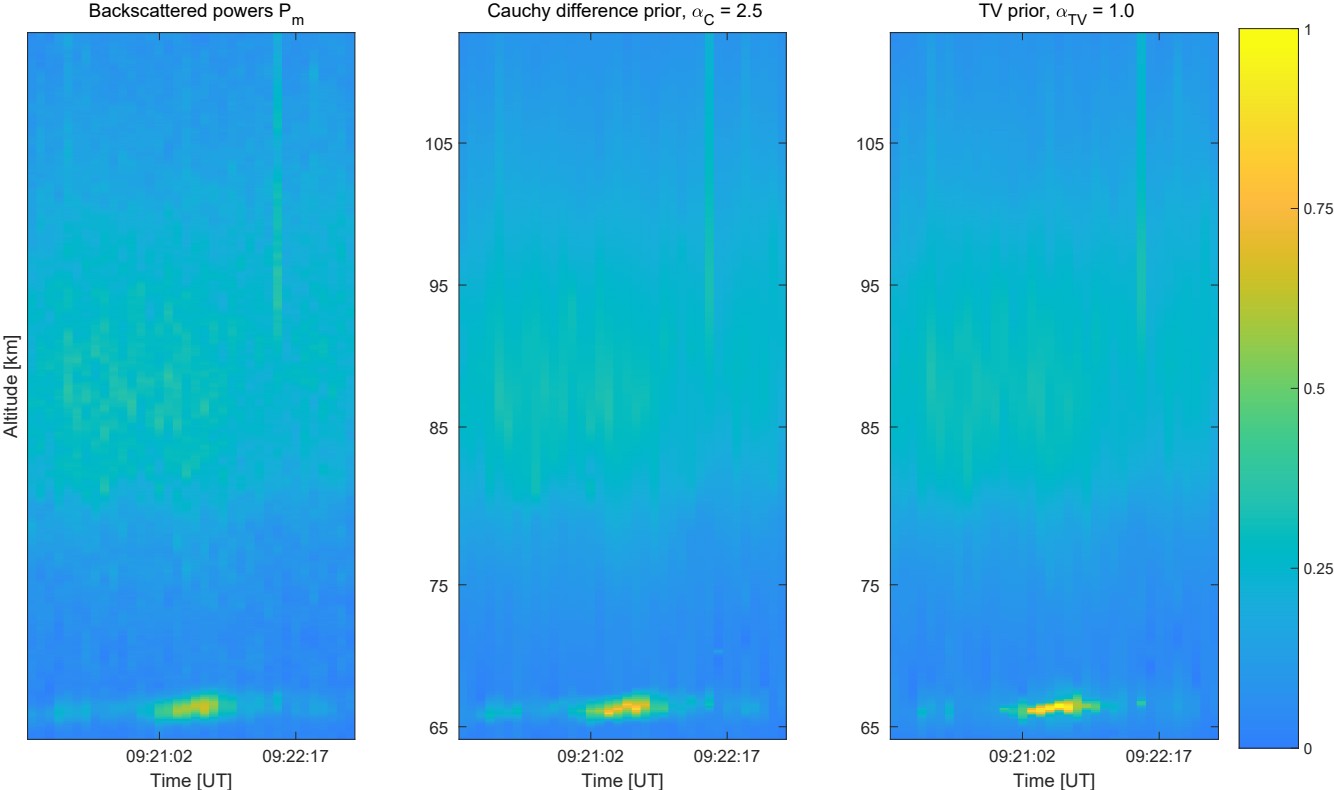

**Figure 7.** PMWE time-series of backscattered power profiles after matched filtering on the left, Cauchy difference prior for $\alpha_C = 2.5$ in the middle, and TV prior with $\alpha_{TV} = 1.0$ on the right

Our analysis of the synthetic data showed that the backscattered power from a sporadic E layer was underestimated, while the width of the layer was overestimated with both priors. Also, the reconstructions with the two prior models were considerably different within the PMWE layer, where a full uncertainty quantification using MCMC might be a useful tool to assess correctness and variability of the obtained estimates. The reconstructions might thus not be accurate within the strong layers, but our results suggest that they are still closer to reality than the power profiles calculated from the matched filter data. We note that further improvements in resolution could be reached if the data were not decoded by the matched filter, and also non-zero time lags of the autocorrelation function were included in the analysis. Such power profiles and covariance matrices could be produced by means of lag profile inversion (Virtanen et al., 2008), after which the prior models could be applied in a post processing step in the same way they were applied to the decoded power profiles in this paper.

In this study we have chosen two process priors, Cauchy difference and TV, which have different parameterisations, thus leading to different estimators. The primary goal herein is that we want both models to detect two distinct parts, the smooth part and the high-frequency part. When comparing the performance of both priors considered the above experiments showcase

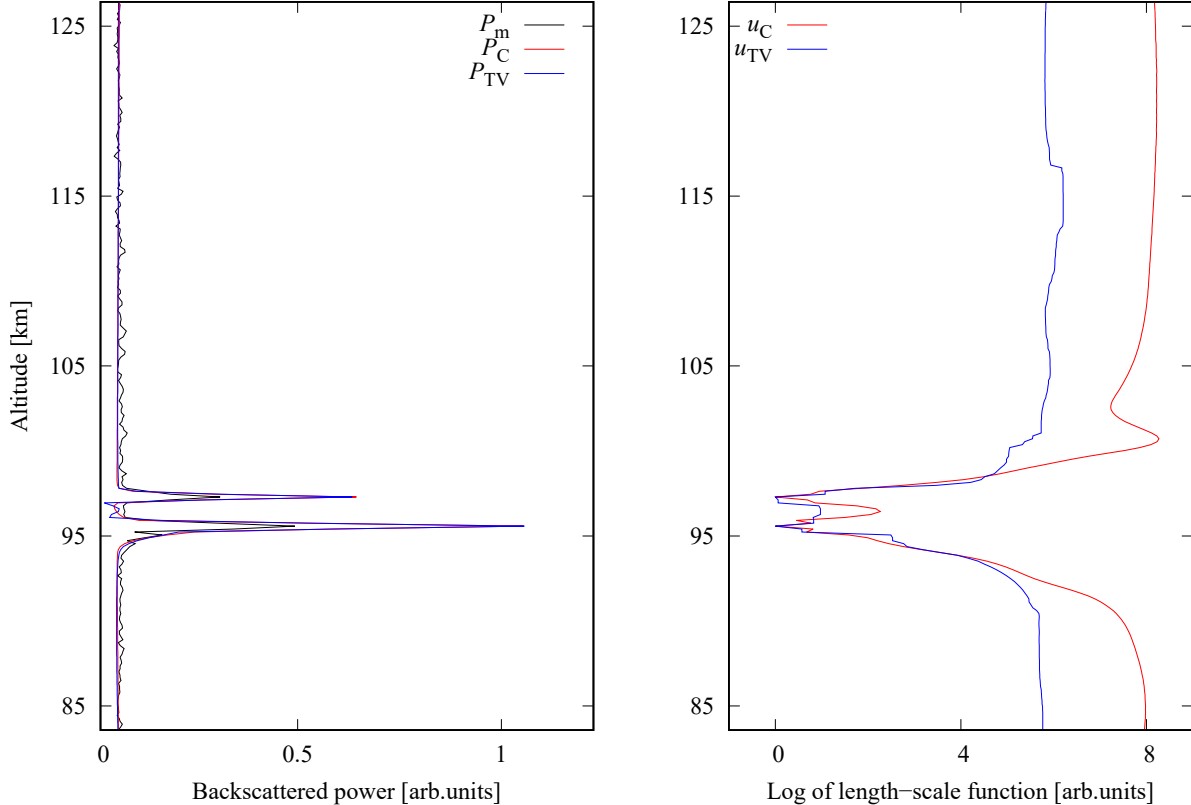

**Figure 8.** PMSE EISCAT data from 12/08/2018, at exact UT time 00:16:40. The left panel shows the comparison between the reconstructions of the unknown signal $\boldsymbol{P}$ with the Cauchy difference prior (red line), with tuning parameter $\alpha_C = 2.5$ and TV (blue line) prior, with $\alpha_{TV} = 5.0$. The backscattered power profile after matched filtering $P_m$ are plotted with black line. The right panel shows corresponding values of the length-scale functions in logarithmic scale.

a slightly different behaviour. The TV prior generally shows good performance to recover higher values for the backscattered power in the narrow profiles compared to the Cauchy difference prior, but this performance is much more tied to an optimal choice of the tuning parameter, ranging from 1 to 5 for TV, whereas the Cauchy difference prior showed good performance in a range of 2.5 to 3 and as such is much more stable and reliable. This different performance is most apparent in Fig. 7. Here, the TV prior shows again an excellent capability in improving the resolution of the narrow PMWE layer, but shows a

tendency to remove weaker layers with the same parameter. On the other hand the Cauchy difference prior shows a good trade-off in improving the resolution of the narrow layer, while keeping weaker layers intact. Nevertheless, both priors show good performance in preserving the smooth background above 75 km, as discussed above. This suggests, that the TV prior would need a separate choice of the tuning parameter for each profile, whereas the Cauchy difference prior shows a more consistent improvement for a fixed value.

Indeed, Cauchy difference priors lead to marginal distributions which can be unimodal or bimodal, or even multimodal (Markkanen et al., 2019). Bimodality is the key ingredient in designing models with rough features, that is, the edges are modelled with bimodal probability densities. For the TV prior, the edges are modelled via the product of two exponential functions, by which edges have uniform density. This means that the Cauchy difference prior promotes rougher features than the TV prior, and this is, in our understanding, the reason for differences in the reconstructions. Nevertheless, even though the differences between the models may seem to be significant, the resulting backscattered power profiles are of similar appearance, thus we believe that the proposed models are robust against parameter tuning within the presented range.

The choice of optimal tuning parameters is generally problematic and this work described the ideal performance of the method, when tuning parameters ($\alpha_C$ or $\alpha_{TV}$) were set to their best possible values (which can be easily done by grid search for synthetic data). In reality, as we have discussed, one needs to adjust those parameters for each data set separately and in the case of TV, possibly adjust even between profiles. In principle, it is also possible to extend the current Bayesian model by setting own priors for the needed tuning parameters and estimating them simultaneously with the other unknowns. Based on earlier experiments in other applications this could be more easily done in the MAP estimation context for the TV prior by (Park and Casella, 2008; Kärkkäinen and Sillanpää, 2012), which also showed a larger need for separate adjustments. For the Cauchy difference prior, we expect that this requires likely stochastic optimisation within the context of MCMC. Finally, we note that a temporal coupling between layers could be incorporated to promote regularity for the measured time-series as investigated by Arjas et al. (2020a), this need is clearly seen in Fig. 7. Nevertheless, this would increase computational complexity significantly.

Finally, we note that computational results were achieved on a workstation with an Intel(R) Core(TM) i7-10510U processor running at 2304 MHz and 4 cores. Average reconstruction times for the Cauchy difference prior range between 1 to 3 minutes per profile. For the TV prior this increases to $2-7$ minutes and is dependent on the convergence of the optimisation for the length-scale function. The computation of a full time-series as in Fig. 7 takes about 120 minutes. Alternatively, if the tuning parameter is chosen constant over time, then computation of profiles can be readily parallelised.

## 6.1 Limitations of the presented model

In the radar signal model presented in Section 2, the incoherent scatter self-noise contribution was neglected and the measurement noise was assumed to be stationary, zero-mean, Gaussian white noise. While this is a reasonable starting point for the analysis technique development, the self-noise contribution in our data may be significant due to the presence of strong layers. The self-noise makes the noise process non-stationary and correlated, which means that one should estimate the full measurement error covariance and use it in the deconvolution process. One should thus consider possibilities to include the self-noise in the signal model and to use the improved model in the hierarchical deconvolution process.

If time resolution of the data analysis is much coarser than duration of a radar code cycle, several observations of the echoes from each code are available, and one can readily calculate the full error covariance matrix of the measurements $\mathbf{P}_m$ with the cost of increased computational complexity (Huuskonen and Lehtinen, 1996). The technique fails at the limit of very long code cycles or very high time resolutions, but this limitation is not specific to our deconvolution technique. Furthermore, the

diagonal of the error covariance can be calculated also for very long code sequences, because the variances do not depend on the phase-coding.

The full measurement error covariance matrix, denoted by $\mathbf{R}$, can be incorporated into the deconvolution model. We can utilise the Cholesky factor $\mathbf{S}$ of $\mathbf{R}$, i.e., $\mathbf{SS}^T = \mathbf{R}$, such that $\mathbf{S}^{-1}\mathbf{P}_m = \mathbf{S}^{-1}\mathbf{AP} + \mathbf{S}^{-1}\varepsilon$. This whitens the error vector, making its components independently distributed. After this, the original algorithm can be used by setting the theory matrix to $\mathbf{A}^* = \mathbf{S}^{-1}\mathbf{A}$ and $\sigma^2 = 1$.

## 7 Conclusion

We have introduced the concept of *hierarchical deconvolution* for incoherent radar scatter data that is capable to estimate the length-scales of the signal from the data itself. This allows for reconstructed backscattered power profiles with varying regularity in presence of narrow layers and smooth background ionization profiles. The recovery is performed in the framework of statistical inversion and estimation of the length-scales is efficiently performed by an optimisation procedure.

In this study we have not investigated other priors than the Cauchy difference and TV prior. Naturally, there is a large selection of priors one can use for specific applications (Gelman et al., 2013): Gaussian priors, geometric priors, and even data-driven priors via neural networks (Adler and Öktem, 2018). The reason why we have concentrated on the Cauchy difference and TV priors is because we want to avoid dyadic structures, specific geometries and dependence of data, which could be well suited for future studies, but out-of-scope here.

The presented results suggest that we can successfully and automatically tune the length-scales of the used prior models in ISR power profile deconvolution, as demonstrated for measured PMWE and PMSE layers. Nevertheless, the obtained results depend on the chosen tuning parameter. Here the presented Cauchy difference prior showed a better performance with similarly chosen parameter values, this suggests it is better suited for the presented deconvolution problem, where many profiles may need to be analysed. We believe that the proposed framework will be be useful in extremely high resolution radar observations and in full-profile ISR plasma parameter analysis.

*Code and data availability.* Codes will be made available after acceptance. The EISCAT data is available for download from the EISCAT web page (www.eiscat.se). For the specific data used in this paper, please contact the authors for details.

*Author contributions.* SR has computed the presented results. AA has developed the reconstruction codes. IV has provided data processing routines. MS, LR, and AH have equally contributed to conceptualisation and design of the study. SR has prepared the manuscript with all authors contributing to the writing.

*Competing interests.* We declare that the authors have no competing interests.

*Acknowledgements.* This work was supported by Academy of Finland Profi 5 (HiDyn) funding for mathematics and AI: data insight for high-dimensional dynamics (grant 326291), and by Profi 2 funding (grant 301542), and Academy of Finland projects 336787, 336796, 338408. We acknowledge the EISCAT Association for the incoherent scatter radar data used in this study. EISCAT is an international association supported by research organizations in China (CRIRP), Finland (SA), Japan (NIPR and ISEE), Norway (NFR), Sweden (VR), and the United Kingdom (UKRI).

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
