# Peer review of "Hierarchical Deconvolution for Incoherent Scatter Radar Data"

_Atmospheric Measurement Techniques, 2021_

## Author Response (AR1)

**Response to Reviewers' Comments**

We thank the reviewers for their detailed reading of the paper, for catching some errors, and for the helpful suggestions for improvement. Please see the following pages for a detailed response and a summary of our changes.

Sincerely,

Snizhana Ross, Arttu Arjas, Ilkka I. Virtanen, Mikko J. Sillanpää, Lassi Roininen, and Andreas Hauptmann 19 January 2021

**Response to Reviewer 1**

I found the description of this hierarchical convolution technique to be clear and wellorganized, and I have high confidence that I could implement the technique based on reading the paper. I think this is an exciting area of development for processing radar data and, in particular, incoherent scatter radar data, and I look forward to future developments. I have some specific comments that follow, but they mainly touch on areas where I think additional information or clarification would improve the paper.

- **Response:** We thank the reviewer for the encouraging evaluation and for excellent suggestions to improve the clarity of the paper. In the following we outline our specific responses to the raised points.
  - 1. In the paragraph containing Equation (7), it is introduced as, "In order to reach resolutions better than the elementary pulse length". I found this slightly confusing on the first read-through because I initially failed to recognize that Equation (7) is a discretization of Equation (5) since my attention had been directed to the resolution issue. The quoted clause implied to me that the form of the following Equation (7) is specialized in order to achieve increased resolution, but in truth the equation would look similar in all cases and is necessary just for discretization. I suggest removing the quoted clause and placing discussion of how to achieve resolutions better than the elementary pulse length to after the description of Equation (7).
    - **Response:** We agree with the reviewer and will change the text before Equation (7) as follows: "In order to reach resolutions better than the elementary pulse length,  $\rightarrow$  In order to move from the continuous time signals to discrete samples,"

We will then add the sentence after Equation (7): "In order to reach resolutions better than the elementary pulse length, we oversample the signal, i.e. use sampling steps shorter than the elementary pulse length."

**Changes:** We have changed the wording as follows.

In line 127: In order to move from the continuous time signals to discrete samples

In line 136, 137: In order to reach resolutions better than the elementary pulse length, we oversample the signal, i.e. use sampling steps shorter than the elementary pulse length.

- 2. Using a mean of 0 for the Gaussian Process prior for P is described as a "convenience", and I appreciate from my own experience with GPs that it is indeed such. Are there other justifications you can provide for why that is an appropriate assumption in this case?
  - **Response:** Our target is in estimating the variability, and especially the high frequency parts (non-stationarities) of P. Our approach is in modelling this variability via the non-stationary covariance function with zero-mean GPs. Alternatively, we could choose a non-zero mean parameter or a continuous-parameter profile. These we could also estimate within our model. However, if we have a continuous-parameter profile, this could lead to overparameterisation and unidentifiability of the unknown objects, as the high-frequency parts would be both in the non-stationary covariance, as well as in the estimated continuous-parameter mean. A straightforward alternative would be to use some other measurements for mean estimation (this has been done e.g. for ionospheric tomography by using ionosonde measurements for mean estimation). As ISR can be considered as the baseline measurement for ionosphere, this is pretty much impossible to achieve with standard experiments (perhaps excluding rocket experiments).

In summary, the zero-mean choice is a rough simplification, but as our target is to detect the non-stationarities and providing general purpose tools (rather than tuning the model for specific cases), we feel that making a more complex choice is not needed for the purpose in this paper. We will add a short comment accordingly after introducing the 0 mean in Section 3.1 and remove the wording "convenience".

- **Changes:** Starting from line 165, we have removed "Convenience" and added the short reasoning: "which is a simplified assumption, but sufficient to detect the desired non-stationary features in this study."
- 3. Similarly, can you provide additional justification for why a Matérn covariance with  $\nu = 1.5$  is chosen? Including a quick statement in the text will help readers who are less familiar with Gaussian Processes so they don't have to turn to one of the references to find the answer.
  - **Response:** We agree that this needs more explanation. The parameter  $\nu$  determines the smoothness of the underlying process. In the case  $\nu = 1.5$ , the process is once mean square differentiable. We choose  $\nu = p+0.5$ ,  $p \in$

 $\mathbb{N}$ , as this provides a Markov approximation for the model, and thus there exists a simple form for the covariance function via stochastic differential equations. Hence by constructing and by choosing  $\nu = 1.5$ , the square-root of the inverted covariance matrix has a tridiagonal structure – which is numerically convenient.

We will add an explanation accordingly to the text.

- **Changes:** We have added the text in lines 172-174: "We choose  $\nu = p + 0.5$ ,  $p \in \mathbb{N}$ , as this provides a Markov approximation for the model. By the construction, the square-root of the inverted covariance matrix has a tridiagonal structure which is numerically convenient".
- 4. It is noted that  $L_l$  is a tridiagonal matrix with reference to Roininen et al. 2014. I suggest adding a quick statement saying why this is the case (e.g. finite differences approximating the derivative) and why it is useful (e.g. efficient computation especially as the problem size scales up). Providing an explicit expression of  $L_l$  as a function of  $l_i$  here would also be good for clarity, although I do note that it appears in-text later in lines 204 and 205.
  - **Response:** This is correct, and as pointed above, the Markov approximation leads to numerically and computationally useful presentation. Moreover, this also allows us to model  $\ell_i$  via increments, thus simplifying both the model and computations.

We will add a short motivation where it is first mentioned and point the reader to the following section for the explicit representation.

- **Changes:** We added text in line 182-184: "[...], motivated by the Markov approximation that leads to a computationally useful presentation. Moreover, this also allows us to model  $\ell$  via increments, thus simplifying both the model and computations. The explicit representation of  $\mathbf{L}_{\ell}$  is given in the Section 4."
- 5. The Figure 3 labels and text discussing the figure refer to u as the "lengthscale function". I think it would be clearer to note that this is the log of the underlying length scale, so that statements like "by factor 3-5 large in smooth parts of the profile" can more easily be associated with the log scale under discussion. Better yet would be to reference the physical units associated with the underlying length scale values.

- **Response:** We agree and will change the figure captions. We also point to our response to Reviewer 2 in Technical corrections 4, that we will add [arb.units] as this depends on the sampling accuracy of the profiles to be recovered.
- **Changes:** Accordingly, the labels to the Figures 3, 4, 6, 8 were changed from "Length-scale function" to "Log of length-scale function [arb. units]"
- 6. The alpha tuning parameters were optimized to minimize the mean squared error between P and  $\hat{P}$ , and the resulting estimates all underestimate the peak power of the sporadic E layer. Presumably this is because the length scale would need to reach a smaller value at those altitudes in order to permit the large gradient that exists there. Did you test higher values for the alpha parameters, and does that end up fitting the sporadic E peak better? What does that do for the quality of the estimates at other altitudes for the background ionosphere? In other words, if one was more interested in the highest quality estimates of either a narrow feature or the background ionosphere at the expense of the other, how does that affect the decision for setting the alpha parameters?
  - **Response:** This is a very good point, that we missed to discuss. We have conducted some more experiments to test if a higher peak could be reached with more parameter tuning. In fact, the particular values for  $\alpha_C$  and  $\alpha_{TV}$  are already (very) close to optimal for the sporadic E peak. Tuning the  $\alpha$  parameters for the estimation of the narrow feature of the peak will primarily affect the outer layers and does not improve the reconstruction of the highest peak power. This nicely underlines the benefit of the length-scale function in the estimation procedure, as it is robust for the narrow layers and the tuning parameters affect primarily the desired smoothness of the outer layers.

We will add a comment to Section 5.1. as follows: "The presented modelling proves to be robust in recovering the peak power. Specifically, choosing different tuning parameters to estimate the narrow feature of the highest peak power not increasing significantly the quality of the reconstruction in comparison to the optimal values of  $\alpha_{\rm C}$  and  $\alpha_{\rm TV}$ . This underlines the need for an adaptive length-scale function in the estimation procedure."

**Changes:** We have added the following to line 260-263: "The presented modelling proves to be robust in recovering the peak power. Specifically, choosing different tuning parameters to estimate the narrow feature of the highest peak power not increasing significantly the quality of the reconstruction in comparison to the optimal values of  $\alpha_{\rm C}$  and  $\alpha_{\rm TV}$ . This underlines the need for an adaptive length-scale function in the estimation procedure."

- 7. Following on from the previous comment: did you test any other prior distributions (i.e. not Cauchy or TV/Laplace) for the length scale difference that might be better suited to really sharp gradients? If not, can you point to directions for future work in this area?
  - **Response:** This is a good point for discussion, in our study we have not chosen any other priors. Naturally, there is a large selection of different priors one can use for specific applications: Gaussian priors (easy to use for continuous models, but not good for rough features), Besov priors (good for rough features, but have dyadic structures due to wavelets), geometric priors (requires model-specific constructions), and data-driven priors via neural networks, especially GANs (requires training data).

As mentioned, for this particular study, we concentrated on the Cauchy and TV priors. This is because we want to avoid the dyadic structures, specific geometries and dependence of data, which could be well suited for future studies, but out-of-scope here. For instance, one could use rougher features by using general alpha-stable processes, but losing the analytical properties of Cauchy distributions, which would further complicate the process due the needed computations of approximations of probability density functions of alpha-stable densities. Moreover, as the Cauchy probability density function is already an infinite-variance model, we suspect that the result of going to rougher models would have a marginal effect to the first-layer GP non-stationary model.

We will add a comment accordingly to the Discussion in Section 6 and point to possible improvements in future studies.

**Changes**: We have added a short discussion of possible other priors to the Conclusions as a possible direction for future studies (Lines 413-417): "In this study we have not investigated other priors than the Cauchy difference and TV prior. Naturally, there is a large selection of priors one can use for specific applications (Gelman et al., 2013): Gaussian priors, geometric priors, and even data-driven priors via neural networks (Adler and Öktem, 2018). The reason why we have concentrated on the Cauchy difference and TV priors, is because we want to avoid dyadic structures, specific geometries and dependence of data, which could be well suited for future studies, but out-of-scope here."

- 8. How specifically did you choose the tuning parameter values for the PMWE and PMSE results? (i.e. What "performance" [line 296] is being optimized?)
  - **Response**: Naturally, the ground truth is not available for the PMWE and PMSE measured signals. As such, the tuning parameters in this case were chosen empirically and were validated visually by professional judgment to improve reconstruction characteristics. Specifically, concentrating on the resolution of the narrow layer and continuity over time (in Figure 7), while maintaining smooth characteristics of the outer layers.

We will add a comment in Sections 5.2 and 5.3 accordingly.

**Changes:** We added the corresponding text to lines 315-318: "The ground truth is not available for the PMWE and PMSE measured signals. As such, the tuning parameters in this case were chosen empirically and were validated visually by professional judgment to improve reconstruction characteristics. Specifically, concentrating on the resolution of the narrow layer and continuity over time (in Figure 7), while maintaining smooth characteristics of the outer layers."

**Response to Reviewer 1. Technical corrections**

1. (line 313) "from in TV prior"  $\rightarrow$  "from the TV prior"? (line 334) "Cauchy difference TV"  $\rightarrow$  "Cauchy and difference TV"?

**Response**: Thank you for carefully reading the manuscript, we will correct the errors.

**Changes:** All the errors have been corrected.

**Response to Reviewer 2**

1. While the model and the algorithm are quite complicated, the authors made a large effort to make them as clear as possible in the text. In addition, no serious errors and flaws were found. On the other hand, it is very hard to overview the whole structure of the model at a glance. For the sake of readers, I give some minor comments in what follows.

The current version of Fig. 2 simply shows the relationship between the parameters, and the model structure is described in detail part by part throughout the sections 2-4. However, the current structure requires readers to go back and forth in the text until the model is understood and this is rather painful. In my opinion, Fig. 2, or perhaps better to add another figure, should also include the model structure itself to grasp the whole structure at a glance. More specifically, it should illustrate relationship of the Gaussian Process and Matern covariance, the additive epsilon and Gaussian pdf and so on in the diagram, as well as MCMC and MAP.

- **Response**: Thank you for the encouraging evaluation and the critical comments. We agree that the structure might be a bit hard to follow at first and welcome the suggestions how to make it more accessible. We have discussed how to best address the reviewers comments to improve the presentation and came the the conclusion that we will include an additional subsection 4.1 on "Model overview", that aids to summarise the whole process. In particular, to address the suggestion of adding another Figure, we will include a flowchart that shows which steps have to be taken and how these relate to each other.
- **Changes:** We have now added a new section 4.1 on "Model overview". We have decided not to include a flow chart, in contrast to what we wrote in the previous response as we felt it will be redundant. Nevertheless, we have now formulated the full hierarchical model in equation (16) followed by a pseudo-code for both, the MAP estimate and MCMC inference. We believe this should help the reader to follow the necessary steps, while all information is now collected in one concise subsection.
- 2. In addition to the logical relationship of the model parts mentioned above, it is recommended if possible schematically to show the sequence (in time) of the procedures to show which part of the model and how to start the calculation from.

- **Response**: Following the previous point, we will not only add the flowchart as visual illustration, but also include a summarising pseudo-code for the MAP and MCMC estimation. We believe that this should provide the necessary overview of the model in a concise manner.
- **Changes:** As previously outline, we have now included a pseudo-code in section 4.1.
- 3. In L.152, which is "Here  $p(P_m|P, L)$  is the likelihood..." (L is intentionally capitalized for readability purpose in this communication), can  $p(P_m|P, L)$  be  $p(P_m|P)$ ? It is because  $P_m$  is presumably conditionally independent from L given P.
  - **Response:** This is correct, we will add a short comment that it can be also simply  $p(P_m|P)$  due to conditional independence.
  - **Changes:** Text added in lines 153, 154: "and due to conditional independence is just  $p(P_m|P)$ "
- 4. Equation (12) indicates the name of prior PDFs (Cauchy & Laplace) but does not show their mathematical forms. While this is accepted in case actual expressions are not concerned, it is recommended to show them in this paper because the definitions of  $\alpha_{C/TV}$  are needed in the following discussions.
  - **Response:** We fully agree on this point and the corresponding mathematical expressions will be added add the corresponding location in Section 3.2. after Equation (12).
  - **Changes:** Lines 188-190: Text added "The probability density functions of Cauchy and Laplace distributions are given as  $p(x) \propto ((x x_0)^2 + s^2)^{-1}$  and  $p(x) \propto \exp(|x x_0|s^{-1})$ , respectively, where  $x_0$  is the center and s is the scaling of the distribution."
- 5. In Figure 5, what is the reason by which the sidelobe of the left plot (PMWE) is wavy while the other (PMSE) is quite smooth?
  - **Response:** We agree, that it is beneficial to include more detail about the sidelobe behaviour. We will add some more explanation in Section 5.2.

Specifically, for the wavy sidelobes we will add: "Considerable side lobes are produced in matched filtering of the shortcodes that are not Barker codes".

Regarding the PMSE, we will add that "the range side lobes are smooth because the long codes behave reasonably well in matched filtering, and sidelobe patterns of each code in the long code sequence are different."

Finally, the corresponding caption to Figure 5 will be changed to: "The pair of 10-bit codes used in the PMWE observation produces significant side lobes in matched filter decoding, while the sequence of 128 61-bit codes used in the PMSE observation leads to a smooth pattern of smaller side lobes"

**Changes**: We have made the following changes.

Line 311, 312: Text added "Considerable side lobes are produced in matched filtering of the shortcodes that are not Barker codes"

Lines 344, 345 : Text added "the range side lobes are smooth because the long codes behave reasonably well in matched filtering, and sidelobe patterns of each code in the long code sequence are different."

Caption to Figure 5, text added "The pair of 10-bit codes used in the PMWE observation produces significant side lobes in matched filter decoding, while the sequence of 128 61-bit codes used in the PMSE observation leads to a smooth pattern of smaller side lobes"

- 6. In Figures 6 and 8, what is the reason by which  $u_C$  and  $u_T V$  are quite different where they are higher than 4.0?
  - **Response**: We acknowledge that this needed more explanation. The fundamental reason is that the Cauchy and TV process priors are different models, and have different parameterisations, thus leading to different estimators. The crucial point is that we want both models to detect two distinct parts, the smooth part and the high-frequency part, that is, even though the estimators look different, both of them lead to estimators which clearly model the targets as wanted. Even though the differences between the models may seem to be significant, the resulting backscattered power profiles are similar, thus one can claim that the model is robust against parameter tuning within this range.

We will add corresponding comments to the Discussion in Section 6. Following Reviewer 1, comment 7, as well as the next point.

- **Changes:** We have added a short comment in the discussion starting line 373: "In this study we have chosen two process priors, Cauchy difference and TV, which have different parameterisations, thus leading to different estimators. The primary goal herein is that we want both models to detect two distinct parts, the smooth part and the high-frequency part." Additionally, we have added a new paragraph to the discussion, starting in line 386: "Indeed, Cauchy difference priors lead to marginal distributions which are either unimodal or bimodal (See Markkanen et al. 2019). Bimodality is the key ingredient in designing models with rough features, that is, the edges are modelled with bimodal probability densities. For the TV prior, the edges are modelled via the product of two exponential functions, by which edges have uniform density. This means that the Cauchy difference prior promotes rougher features than the TV prior, and this is, in our understanding, the reason for differences in the reconstructions. Nevertheless, even though the differences between the models may seem to be significant, the resulting backscattered power profiles are of similar appearance, thus we believe that the proposed models are robust against parameter tuning within the presented range."
- 7. On p. 17, the authors discuss the difference between the results from Cauchy and Laplace priors, but its underlying reason is not mentioned at all. Since the difference is very curious and interesting, it is preferable to mention some of your ideas about it if you have any.
  - **Response**: This nicely complements the previously raised points. Indeed, our main idea is that the Cauchy process priors lead to marginal distributions which are either unimodal or bimodal (See Markkanen et al. 2019). Bimodality is the key ingredient in building models with rough features, that is, the edges are modelled with bimodal probability densities. For the TV prior, the edges are de facto modelled via the product of two exponential functions, which means, that at the edges there is "uniform" density. This means that the Cauchy process prior promotes rougher features than the TV, and this is, in our understanding, the reason for differences in the reconstructions.

As previously mentioned, we will add more details on priors and their differences to the Discussion in Section 6.

**Changes:** Please see response to the last point 6.

**Response to Reviewer 2. Technical corrections**

- 1. L169 and L177: Roininen et al. (2014) corresponds to two papers in the reference list. Please identify which one it is.
  - **Response:** Thank you, the appropriate publications will be identified correctly in the text.
  - **Changes:** Lines 171, 181. The appropriate publication Roininen et al. (2014b) has been identified
- 2. L169: Is "partial differential equation" correct? (10) and (11) look like ordinary differential equations.
  - **Response**: As it depends on dimensionality, we will remove "partial" to avoid any ambiguity.

Changes: Line 171, "partial" was removed

3. L313: out from  $in \rightarrow out$  in L334: difference  $TV \rightarrow difference$  and TVL395:  $STEL \rightarrow ISEE$

**Response**: Thank you for your careful reading, the errors will be corrected.

**Changes:** All the errors have been fixed

- 4. Figures 3, 4, 6, & 8: Is the "unit" of length-scale function [km] or [log km]?
  - **Response:** We agree with the reviewer that on the Figures 3, 4, 6 and 8 units of the log length-scale functions needs to be clarified. Following Reviewer 1 comment 5, we will change the caption to "logarithm of length-scale function". The units are more difficult, as they are non-physical and depend on the sampling resolution of the underlying profile and are assumed to be universally 1. Thus, we have decided to add [arb.units] here as well.
  - **Changes:** Labels to the Figures 3, 4, 6, 8 were changed from "Length-scale function" to "Log of length-scale function [arb. units]"

---

## Referee Report (RR1)

This manuscript presents a novel approach to radar data analysis by applying recently developed mathematical techniques involving hierarchical statistical models and hyperpriors. The manuscript clearly motivates the utility of these techniques for radar data analysis in situations where different atmospheric targets are present with significantly different length-scales, and no single characteristic length scale can reasonably be assumed a priori. This situation is common in radar studies of the D- and E-region ionosphere where sporadic E layers, PMSE, or PMWE can be observed. Overall the results presented in this manuscript are a promising proof-of-concept demonstrating the utility of these hierarchical techniques. Nonetheless, the models used make certain inaccurate assumptions about radar signals that make the present work incomplete. The limitations of these assumptions require explanation and discussion of how future work could apply these techniques to more realistic signal models.

**Major** Issues**

1. The use of a constant and diagonal noise covariance matrix is not correct for the radar signals of interest. This study assumes that the measurements  $P_m$  are normally distributed with covariance matrix  $\sigma^2 \mathbf{I}$ , where  $\sigma^2$  is a known constant. This assumption is only appropriate for weak signals in a particular limit, and it will generally not be appropriate for strong signals such as PMSE and PMWE. Line 136 acknowledges that the "self-noise" contribution from the target may violate this assumption in some cases without adequate additional discussion.

The correct way to model the radar signals is to write Eq. 4 as

$$z_m(t) = (W * \sigma)(t) + n(t)$$

where both the target scattering amplitudes  $\sigma$  and the noise contributions n are independent Gaussian random processes. Assuming the noise power,  $N = E\left\{|n|^2\right\}$  is independently known, Eq. 5 should be written as

$$P_m(t) = \frac{1}{M} \sum_{\ell=1}^{M} |z_m^{\ell}|^2 - N.$$

In general  $P_m(t)$  is not Gaussian, but if M is sufficiently large one may invoke the central limit theorem and derive an approximate Gaussian distribution for  $P_m(t)$ .

If the target signals are extremely weak compared to the receiver noise, then the covariance matrix of  $P_m$  is simply  $\frac{N^2}{M}\mathbf{I}$ . Therefore the model from this manuscript is correct in this weak signal limit if one identifies  $\sigma^2 = \frac{N^2}{M}$ . Many of the signals of interest for this work, such as sporadic E, PMSE, and PMWE will usually not satisfy this weak signal limit, and therefore the model in this manuscript is inappropriate.

In the high signal limit, the complete expression for the relevant covariance matrix of  $P_m(t)$  has all of the following difficult properties

• It is not a constant

- It is non-diagonal for every point-spread function other than the ideal Dirac delta (self-clutter effect).
- It explicitly depends on the signal power  $P = E\left\{|\sigma(t)|^2\right\}$ , which is unknown a priori (self-noise effect).
- It generally depends on the pulse-to-pulse correlation function of the target as well,  $R_{\ell,k} = E\left\{\sigma^{\ell}(t)\bar{\sigma}^{k}(t)\right\}$ , which is also unknown a priori.

For interpulse periods of several milliseconds the pulse-to-pulse correlations can be neglected for normal E-region incoherent scatter and for sporadic E layers. For D-region incoherent scatter, PMWE, and PMSE, however, these correlations are significant, and the individual  $\sigma^{\ell}$  from different pulses cannot be analyzed as independent measurements.

A complete formulation that correctly treats the complete covariance matrix is probably best left to future work, but the manuscript should at least discuss whether the method could conceivably accommodate more accurate treatments of the covariance matrix in the future.

- 2. The manuscript does not discuss whether the estimation scheme could accommodate selfnoise effects. Equations 13, 14, and 15 are independent of the unknown P if  $\sigma^2$  is assumed to be known. If self-noise effects are included, however, then the data covariance depends on the unknown powers P, and these three equations cannot be solved. The manuscript should discuss strategies for dealing with this difficulty. One possibility is to use  $P_m$  instead of P when evaluating the self-noise contributions. Another possibility is an iterative approach where  $\hat{P}$  from the previous iteration is used to evaluate the self-noise contributions for the next iteration.
- 3. The manuscript does not explain how the data variances are set for the examples. Lines 247-250 describe a synthetic signal generation process that will produce realistic radar signals with self-noise and self-clutter included. As explained above these signals will be inconsistent with a constant  $\sigma^2$ . The real EISCAT signals will also contain self-noise and self-clutter that are inconsistent with a constant  $\sigma^2$ . The manuscript does not explain what value is used for  $\sigma^2$  when inverting these example signals, and the results will likely depend on the choice of  $\sigma^2$ .
- 4. The prior model for P does not constrain the solution to be positive. The scattering power is always a positive number, and it is physically related to quantities that are positive by definition (e.g. electron density). Nonetheless, the prior model for P discussed in section 3.1 is a zero-mean Gaussian process, which implies that negative numbers are equally as likely as positive numbers, a priori. The negative numbers are unphysical. The manuscript should discuss why this prior was chosen and whether the technique could be adapted to use more physical priors in the future.
- 5. The use of arbitrary units power units throughout the examples limits the reader's ability to assess the signal-to-noise regime. While arbitrary units are acceptable, the manuscript should state the noise power level in the same arbitrary units and state the number of samples M involved. As presented it is impossible to determine the signal-to-noise ratios of the signals and how large the self-noise and self-clutter effects are likely to be.

**Minor Corrections**

1. Line 258 should read "explicitly control"

---

## Author Response (AR2)

**Response to Reviewers' Comments**

We thank the reviewers for their detailed reading of the paper, for catching some errors, and for the helpful suggestions for improvement. Please see the following pages for a detailed response and a summary of our changes.

Sincerely,

Snizhana Ross, Arttu Arjas, Ilkka I. Virtanen, Mikko J. Sillanpää, Lassi Roininen, and Andreas Hauptmann May 2, 2022

**Response to Editor**

As you can see from the reviews, your paper requires minor revision before it is ready for publication. It is not expected that you redo the whole method to include the high SNR cases. You might want to leave that for a future effort. However, some discussions and limitations of the current method, need to be included (see reviewer 1's suggestions).

**Response:** We thank the editor for the possibility to address the further comments and for the generous time extension for the revision. As suggested we have added a section 6.1 on "Limitations of the presented model" where the mentioned issues are discussed and added further information to the manuscript were necessary.

**Response to Reviewer 1**

This manuscript presents a novel approach to radar data analysis by applying recently developed mathematical techniques involving hierarchical statistical models and hyperpriors. The manuscript clearly motivates the utility of these techniques for radar data analysis in situations where different atmospheric targets are present with significantly different length-scales, and no single characteristic length scale can reasonably be assumed a priori. This situation is common in radar studies of the D- and E-region ionosphere where sporadic E layers, PMSE, or PMWE can be observed. Overall the results presented in this manuscript are a promising proof-ofconcept demonstrating the utility of these hierarchical techniques. Nonetheless, the models used make certain inaccurate assumptions about radar signals that make the present work incomplete. The limitations of these assumptions require explanation and discussion of how future work could apply these techniques to more realistic signal models.

1. The use of a constant and diagonal noise covariance matrix is not correct for the radar signals of interest. This study assumes that the measurements  $P_m$  are normally distributed with covariance matrix  $\sigma^2 \mathbf{I}$ , where  $\sigma^2$  is a known constant. This assumption is only appropriate for weak signals in a particular limit, and it will generally not be appropriate for strong signals such as PMSE and PMWE. Line 136 acknowledges that the "self-noise" contribution from the target may violate this assumption in some cases without adequate additional discussion.

The correct way to model the radar signals is to write Eq. 4 as

$$z_m(t) = (W * \sigma)(t) + n(t)$$

where both the target scattering amplitudes  $\sigma$  and the noise contributions nare independent Gaussian random processes. Assuming the noise power,  $N = E\{|n|^2\}N$  is independently known, Eq. 5 should be written as

$$P_m(t) = \frac{1}{M} \sum_{\ell=1}^{M} |z_m^{\ell}|^2 - N$$

In general  $P_m(t)$  is not Gaussian, but if M is sufficiently large one may invoke the central limit theorem and derive an approximate Gaussian distribution for  $P_m(t)$ . If the target signals are extremely weak compared to the receiver noise, then the covariance matrix of  $P_m$  is simply  $\frac{N^2}{N}\mathbf{I}$ . Therefore the model from this manuscript is correct in this weak signal limit if one identifies  $\sigma^2 = \frac{N^2}{M}$ . Many of the signals of interest for this work, such as sporadic E, PMSE, and PMWE will usually not satisfy this weak signal limit, and therefore the model in this manuscript is inappropriate.

In the high signal limit, the complete expression for the relevant covariance matrix of  $P_m(t)$  has all of the following difficult properties

- It is not a constant
- It is non-diagonal for every point-spread function other than the ideal Dirac delta (self-clutter effect).
- It explicitly depends on the signal power  $P = E\{|\sigma(t)|^2\}$ , which is unknown apriori (self-noise effect).
- It generally depends on the pulse-to-pulse correlation function of the target as well,  $R_{\ell,k} = E\left\{\sigma^{\ell}(t)\bar{\sigma}^{k}(t)\right\}$ , which is also unknown apriori.

For interpulse periods of several milliseconds the pulse-to-pulse correlations can be neglected for normal E-region incoherent scatter and for sporadic E layers. For D-region incoherent scatter, PMWE, and PMSE, however, these correlations are significant, and the individual  $\sigma^{\ell}$  from different pulses cannot be analyzed as independent measurements.

A complete formulation that correctly treats the complete covariance matrix is probably best left to future work, but the manuscript should at least discuss whether the method could conceivably accommodate more accurate treatments of the covariance matrix in the future.

- **Response:** We agree that the high SNR around the strong layers makes the measurement variances range dependent and probably also leads to measurement errors correlations. However, the simple model with a constant variance  $\sigma^2$  is a practical choice for this manuscript, in which the main emphasis is in the range-dependent length-scales. The full measurement error covariance matrix could be calculated from the data if samples from each individual radar pulse were stored separately and if the radar code cycle is not excessively long. The latter limitation is because the covariance structure is different for each code in a cycle.
- **Changes:** We have added Section 6.1. where we discuss the error covariance calculation:

In the radar signal model presented in Section 2, the incoherent scatter self-noise contribution was neglected and the measurement noise was assumed to be stationary, zero-mean, Gaussian white noise. While this is a reasonable starting point for the analysis technique development, the selfnoise contribution in our data may be significant due to the presence of strong layers. The self-noise makes the noise process non-stationary and correlated, which means that one should estimate the full measurement error covariance and use it in the deconvolution process. One should thus consider possibilities to include the self-noise in the signal model and to use the improved model in the hierarchical deconvolution process.

If time resolution of the data analysis is much coarser than duration of a radar code cycle, several observations of the echoes from each code are available, and one can readily calculate the full error covariance matrix of the measurements  $\mathbf{P}_m$  with the cost of increased computational complexity (Huuskonen and Lehtinen, 1996). The technique fails at the limit of very long code cycles or very high time resolutions, but this limitation is not specific to our deconvolution technique. Furthermore, the diagonal of the error covariance can be calculated also for very long code sequences, because the variances do not depend on the phase-coding.

- 2. The manuscript does not discuss whether the estimation scheme could accommodate self- noise effects. Equations 13, 14, and 15 are independent of the unknown P if σ2 is assumed to be known. If self-noise effects are included, however, then the data covariance depends on the unknown powers P, and these three equations cannot be solved. The manuscript should discuss strategies for dealing with this difficulty. One possibility is to use Pm instead of P when evaluating the self-noise contributions. Another possibility is an iterative approach where P̂ from the previous iteration is used to evaluate the self-noise contributions for the next iteration.
  - **Response**: In theory, the measurement error covariance matrix could be used in the inversion via Cholesky factorization, but implementing this is left for a future work.
  - Changes: Section 6.1.:

The full measurement error covariance matrix, denoted by  $\mathbf{R}$ , can be incorporated into the deconvolution model. We can utilise the Cholesky factor  $\mathbf{S}$  of  $\mathbf{R}$ , i.e.,  $\mathbf{SS}^T = \mathbf{R}$ , such that  $\mathbf{S}^{-1}\mathbf{P}_m = \mathbf{S}^{-1}\mathbf{AP} + \mathbf{S}^{-1}\boldsymbol{\epsilon}$ . This whitens the error vector, making its components independently distributed. After this, the original algorithm can be used by setting the theory matrix to  $\mathbf{A}^* = \mathbf{S}^{-1}\mathbf{A}$  and  $\sigma^2 = 1$ .

- 3. The manuscript does not explain how the data variances are set for the examples. Lines 247-250 describe a synthetic signal generation process that will produce realistic radar signals with self-noise and self-clutter included. As explained above these signals will be inconsistent with a constant  $\sigma^2$ . The real EISCAT signals will also contain self-noise and self-clutter that are inconsistent with a constant  $\sigma^2$ . The manuscript does not explain what value is used for  $\sigma^2$  when inverting these example signals, and the results will likely depend on the choice of  $\sigma^2$ .
  - **Response:** We have added the values of the standard deviations (in the same arbitrary units that are used in the figures). We have included also the thermal background noise levels, because the standard deviations used in the inversion were larger than the thermal background to accommodate for the self-noise from the strong layers.

**Changes:** Line 240: a noise variance $\sigma^2$ and**

Line 269: The results depend also on the standard deviation of the measurement error, which was set to  $\sigma = 0.1$  (in the arbitrary units used in Fig. 3). The value is larger than standard deviation of the background noise in the averaged data (0.04) to accommodate for the self-noise from the strong scattering layer.

Line 317: Standard deviation of the measurement error was set to  $\sigma = 0.1$  (in the units used in Fig. 6). The value is an order of magnitude larger than the thermal background noise in the time-averaged data (0.01) to accommodate for the significant self-noise from the strong layer.

Line 355: Measurement error standard deviation was set to  $\sigma = 0.06$  (in the units used in Fig. 8). The value is again considerably larger than the thermal background level (0.005) to accommodate from the self-noise from the very strong layer.

4. The prior model for P does not constrain the solution to be positive. The scattering power is always a positive number, and it is physically related to quantities that are positive by definition (e.g. electron density). Nonetheless, the prior model for P discussed in section 3.1 is a zero-mean Gaussian process, which implies that negative numbers are equally as likely as positive numbers, a priori. The negative numbers are unphysical. The manuscript should discuss why this prior was chosen and whether the technique could be adapted to use more physical priors in the future.

- **Response:** While the prior model as such promotes values which can be negative, and thus non-physical, the estimators produced are dominated by the likelihood which typically guarantees positivity of the estimators. In the case the estimators would be negative, then the algorithm can be considered to produce non-physical estimators, and we can use this information as an indicator to pinpoint the cases where, e.g., the data is somehow corrupt. Naturally we could force the prior to be non-negative with standard tricks, like logarithmic transformation of the unknown. However, this would induce non-linearities and further complicate the computations, and thus increasing the computation times significantly. Thus, even though of the possible negativity of the prior process, in practice, this is computationally faster, and provides clear estimators for all the properly measured cases.
- **Changes:** We added the following test to Section 3.1: "We note that the prior model is a zero-mean process with negative, and thus non-physical values. We could force the prior to be non-negative with standard tricks, like logarithmic transformation of the unknown. However, this would induce non-linearities and thus increase computation times. In addition, as the likelihood typically guarantees positivity of the estimators, so one could consider non-physical estimators as indicator pinpointing the cases where, e.g., the data is somehow corrupt. "
- 5. The use of arbitrary units power units throughout the examples limits the reader's ability to assess the signal-to-noise regime. While arbitrary units are acceptable, the manuscript should state the noise power level in the same arbitrary units and state the number of samples M involved. As presented it is impossible to determine the signal-to-noise ratios of the signals and how large the self-noise and self-clutter effects are likely to be.
  - **Response**: We have added the number of averaged pulses and standard deviation of the background noise for each of the examples.
  - **Changes:** Line 253: average power profiles were calculated over 665 subsequent transmitted pulses, which leads to 1 s time resolution.

Line 313: average profiles of the backscattered power were calculated with 0.9 s (665 pulses) time resolution.

Line 353: The profile is an average over 128 subsequent pulses (0.2 s in time).

Changes related to noise power levels are included in our response to comment 3.

**Minor Corrections**

1. Line 258 should read "explicitly control"

**Response:** We believe that the comment was about line 58, which said "without the need to explicit control" instead of 258.

Changes: Line 58: explicitly control